

# Data Driven Regional Weather Forecasting

Randall Clark, Luke C. Fairbanks [1], Ramon E. Sanchez, Pacharadech
Wacharanan,
Department of Physics
University of California San Diego
La Jolla, CA 92093, USA

Henry D. I. Abarbanel
Department of Physics
and
Marine Physical Laboratory (Scripps Institution of Oceanography)
University of California San Diego
La Jolla, CA 92093,USA
habarbanel@ucsd.edu
  https://orcid.org/0000-0002-4690-6081

November 7, 2022

---

[1]Corresponding Author: lfairban@ucsd.edu



# Contents







# Abstract

Using data alone, without knowledge of underlying physical models, nonlinear discrete time regional forecasting dynamical rules are constructed employing well tested methods from applied mathematics and nonlinear dynamics. Observations of environmental variables such as wind velocity, temperature, pressure, etc allow the development of forecasting rules that predict the future of these variables only. A regional set of observations with appropriate sensors allows one to forgo standard considerations of spatial resolution and uncertainties in the properties of detailed physical models. Present global or regional models require specification of details of physical processes globally or regionally, and the ensuing, often heavy, computational requirements provide information of the time variation of many quantities not of interest locally. In this paper we formulate the construction of data driven forecasting (DDF) models of geophysical processes and demonstrate how this works within the familiar example of a 'global' model of shallow water flow on a mid-latitude $\beta$ plane. A sub-region, where observations are made, of the global flow is selected. A discrete time dynamical forecasting system is constructed from these observations. DDF forecasting accurately predicts the future of observed variables.



# 1 Introduction

In geophysical models for global and regional weather or climate forecasting, solutions of the Navier-Stokes equations, expressing conservation of mass, and momentum, are accompanied by a thermodynamic equation describing energy conservation. An equation of state relating the thermodynamic state variables to each other is required as are further parameterizations to represent unresolved Physics below the model grid scale. Cloud moisture dynamics is a critical example of the latter.

Numerical solutions for these partial differential equations (PDEs) using, for example, a finite difference method Press et al. (2007); Olver (2020) lead to formulations with a very large number of ordinary differential equations (ODEs) at a global set of spatial grid points. In global models the number of these degrees of freedom (ODEs) may range from $10^8$ to $10^{10}$ while for regional models this may still be large, say $10^6$ to $10^7$. Even then the resolution of the largest operational GCMs is today about 9 km in the horizontal Stevens et al. (2019). Models with scales down to 1.4 km are being developed and tested Wedi et al. (2020), and the computational challenges grow as these are realized.

If one's interest is in forecasting the weather or climate only in a selected region, another point of view may be employed. This builds the relevant dynamics using only observations of the state variables (pressure, velocities, temperatures, ...) one wishes to forecast. Knowledge of the forcing of the system at the location of the observations is required, but this is already estimated in the formulation of the global models Roeckner et al. (2003); Staff (2021).

Working from data alone avoids uncertainties in initial conditions for the ODEs, uncertain physical features of the models and their boundary conditions, and the like. It also circumvents the growing computational complexity as the spatial resolution of big models is increased.

Not surprisingly, one loses something in a formulation that bypasses knowledge of the fundamental physical dynamical equations of the problem. At the same time the computations for forecasting observables only is enormously simplified compared to calculating a full set of physical properties in a region or globally.

The subject of this paper is a method, which we call Data Driven Forecasting (DDF). It is a combination of applied mathematical tools that were well developed some time ago Hardy (1971); Micchelli (1986); Broomhead





and Lowe (1988); Schaback (1995); M. D. J. (2002); Buhmann (2009) with
the essential Physics of how one can learn properties of nonlinear dynami-
cal systems through observations of a subset of the dynamical variables of
that system Takens (1981); Eckmann and Ruelle (1985); Sauer et al. (1991);
Abarbanel (1996); Kantz and Schreiber (2004).

## 1.1 Recent Work in Using Machine Learning Methods in Weather and Climate Modeling

The interest in using data driven modeling in earth systems problems is
both intense and productive. It is well documented in a recent theme issue
of the Proceedings of the Royal Society A entitled "Machine Learning for
weather and climate modeling" (The articles can be accessed directly at
www.bit.ly/TransA-2194) Chantry et al. (2021)

Efforts are directed toward replacing large numerical weather forecast-
ing GCMs with data trained networks (machine learning or ML) Shi et al.
(2015) Comrie (1997) Krasnopolsky et al. (2002) Geer (2021) Dueben and
Bauer (2018). While there is success in these investigations, there remains
hesitancy in the climate and weather forecasting community to openly ac-
cept these methods as they are viewed as black boxes that have tenuous
relation to the physics that they model other than the data that is provided
to them Schultz et al. (2021). It is for reasons like this that there have been ef-
forts in ML research to investigate methods with a large focus on the efficacy
and trustworthiness of ML tools Mackowiak et al. (2021) Huang et al. (2020).
The ML community in weather forecasting have gone an additional step fur-
ther by implementing hybrid models that are machine learning devices that
include physical constraints in some form Wandel et al. (2020) Grover et al.
(2015) Daw et al. (2017). There have been studies showing that weather
forecasting tools that account for physical constraints (typically in their loss
functions) show improvement over ML tools that utilize restricted knowl-
edge of the underlying physics Kashinath et al. (2021). In summary, there
is both an interest and benefit to the hybridization of physics and ML mod-
els Watson-Parris (2021); Brajard et al. (2021).

In this paper we follow a complementary but different path in realizing
nonlinear, dynamical forecasting rules based on observed data alone. We
call it data driven forecasting or DDF. We argue that the DDF modeling
strategy we descibe in this paper embodies the hybridization of physics and





ML in a way not typically performed to reap the forecasting benefits while maintaining a high degree of transparency in how it actually operates.

DDF as will be shown in its formulation and explicitly in a simple, familiar geophysical model incorporates both the physics of the underlying model and ML tools to create an update rule for forecasting. Unlike many hybridization ML models, it does not enforce a physics constraint in its training function. The training of the ML model proceeds as standard regularized ridge regression. The update rule for DDF is a sum of radial basis functions and physically inspired terms which include, but are not limited to, forcing terms and polynomial variables in the original model. The separation of the forcing of the intrinsic geophysical fluid dynamics is both a feature of the underlying fluid dynamical equations and makes testing of the generalization to innovative forcing straightforward.

As we use DDF in this paper, a subset of the degrees of freedom of a 'global' model, which for purposes of illustration and simplicity is taken as a one layer shallow water flow on a $\beta$-plane, are considered observed. Then using the familiar method of time delay embedding Takens (1981); Abarbanel (1996), we introduce the physics of the global flow into the regional forecast. It is clear that this paper represents a proof of principle in using DDF and addressing how the method scales to large, increasingly realistic models is still to be accomplished. In using the DDF forecasting formulation on field data there is a requirement for comparison of computational complexity to mainstream GCMs that is an important part of continuing work.

## 1.2   Plan for this Paper

1. To begin we describe the DDF method and its forecasting goals in the context of a simple fluid dynamics problem of geophysical interest: shallow water flow (SWE) Jiang et al. (1994); Pedlosky (1986); Vallis (2017).

2. The discussion of the SWE example illustrates the key ideas in a concrete example and provides us the basis for a general consideration of the problem of learning from observations alone how to forecast the future development of those observations.

3. We then return to the SWE example to show explicitly how the DDF method is implemented and performs.



4. A Summary and Discussion completes the main body of the paper.

5. An Appendix contains many of the details describing how one estimates the parameters in a DDF model. The discussion is more general than the simple geophysical examples in the body of the paper and may be widely useful beyond this paper.

## 2   Shallow Water Flow on a $\beta$ plane used to illustrate DDF

To illustrate the main ideas in this paper we begin with a familiar example which, though simple, illustrates the ideas in a useful context. The example is shallow water flow on a $\beta$ plane, and this is introduced as the *global* model of a geophysical flow. This is two dimensional flow with state or dynamical variables of velocities in the x and y direction and the height of the fluid: $\{v_1(\mathbf{r},t), v_2(\mathbf{r}.t), \zeta(\mathbf{r},t)\} = \{u(\mathbf{r},t), v(\mathbf{r},t), \zeta(\mathbf{r},t)\}$. $\mathbf{r} = [r_1, r_2] = [x, y]$. The partial differential equations of shallow water flow are solved on a two dimensional global grid in a finite difference approximation for the state variables $\{v_1(i,j,t), v_2(i,j,t), \zeta(i,j,t)\}$ with the integers $(i,j)$ denoting locations on the grid.

We then select a subgrid as the region where we record observations of the state variables $\{v_1(I,J,t), v_2(I,J,t), \zeta(I,J,t)\}$ with the integers $(I,J)$ denoting locations on the regional subgrid.

In this paper we address how data from these regional measurements alone, without knowledge of the underlying dynamical equations or knowledge of the states of the global system outside the subgrid, can allow us to forecast the states of the regional variables $\{v_1(I,J,t), v_2(I,J,t), \zeta(I,J,t)\}$.

In applying the methods developed here one needs only measurements of the state variables we wish to forecast at selected spatial locations in a sub-region. There is no 'grid' where we must place the sensors for the desired observations.

We do not require information about the state of the system outside the region. The forcing of the fluid in the selected region is required. That is already in the global formulation of the problem Roeckner et al. (2003); Staff (2021).




## 2.1 Fluid Equations on a Grid; The Global Model

We initiate our considerations with the one layer Shallow Water Equations (SWE) on a mid-latitude $\beta$ plane Jiang et al. (1994); Pedlosky (1986); Vallis (2017). The plane has coordinates $\mathbf{r} = \{r_1, r_2\} = \{x, y\}$, and the dynamical variables of the flow are the velocities in the x and the y directions $\{v_1(\mathbf{r}, t) = u(\mathbf{r}, t), v_2(\mathbf{r}, t) = v(\mathbf{r}, t)\}$ and the fluid height $\zeta(\mathbf{r}, t)$. $\nabla_\perp = (\frac{\partial}{\partial r_1}, \frac{\partial}{\partial r_2})$.

The SWE take the form

$$\frac{\partial \zeta(\mathbf{r}, t)}{\partial t} + \nabla_\perp(\zeta(\mathbf{r}, t)\mathbf{v}(\mathbf{r}, t)) = 0$$

$$\frac{\partial \mathbf{v}(\mathbf{r}, t)}{\partial t} + \mathbf{v}(\mathbf{r}, t) \cdot \nabla_\perp \mathbf{v}(\mathbf{r}, t) + f(\mathbf{r})\hat{z} \times \mathbf{v}(\mathbf{r}, t) = -g\nabla_\perp \zeta(\mathbf{r}, t))$$

$$+A\nabla_\perp^2 \mathbf{v}(\mathbf{r}, t) - \epsilon \mathbf{v}(\mathbf{r}, t) + \mathcal{F}(\mathbf{r}, t). \tag{1}$$

The pressure is given by the hydrostatic relation $p(\mathbf{r}, z, t) = g\rho(\zeta(\mathbf{r}, t) - z); p(\mathbf{r}, \zeta(\mathbf{r}, t), t) = 0$. Body forces driving the fluid are $\mathcal{F}(\mathbf{r}, t) = [\mathcal{F}_1(\mathbf{r}, t), \mathcal{F}_2(\mathbf{r}, t)]$.

We solve the SWE on a $n_x \times n_y$ grid $\{r_1 = r_{10} + i\Delta x, r_2 = r_{20} + j\Delta y\}$; $i = 0, 1, ..., n_x - 1$; $j = 0, 1, ..., n_y - 1$ using, for example, periodic boundary conditions

$$v(0, y, t) = v(L_x, y, t); \ u(x, 0, t) = u(x, L_y, t); \ L_x = n_x\Delta x; \ L_y = n_y\Delta y. \tag{2}$$

The fluid density is constant and chosen to be $\rho = 1 kg/m^3$. $f(\mathbf{r}) = f_0 + \beta r_2$ is the local rotation of the earth, A is an effective kinematic viscosity, $\epsilon$ is a Rayleigh friction coefficient.

The solution on this grid comprises our *global* dynamics. There are $D_G = 3(n_x \times n_y)$ ODEs for the states $\mathbf{S}(i, j, t) = \{v_1(i, j, t), v_2(i, j, t), \zeta(i, j, t)\}$:

$$\frac{d\mathbf{S}(i, j, t)}{dt} = \mathbf{F}_{i,j}(\mathbf{S}(i, j, t), \theta) + [\mathcal{F}(i, j, t), 0]. \tag{3}$$

$\theta$ are fixed parameters in the global system. $\mathbf{F}_{i,j}(\mathbf{S}, \theta)$ is the vector field of the $D_G$ (global) nonlinear differential equations for the shallow water flow.

## 2.2 Dynamics on a Subgrid; The Regional Model

Next select a subgrid, **our region**, where we make measurements. The locations in the observation region are denoted by $\mathbf{R} = \mathbf{R}_0 + \{R_{10} + I\Delta r_1, R_{20} + J\Delta r_2\}$; $I = 0, 1, 2, ..., N_x - 1, J = 0, 1, 2, ..., N_y - 1$; $N_x \leq n_x, N_y \leq n_y$. It is



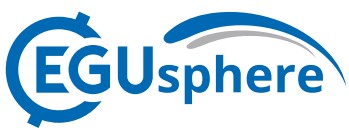

<small>232</small> in this region that we collect observations at a subset of the full complement
<small>233</small> of state variables $\mathbf{S}(\mathbf{r}, t)$.

<small>234</small>    These observations are $\mathbf{O}(\mathbf{R}, t) = \mathbf{O}(I, J, t) = \{v_1(I, J, t), v_2(I, J, t), \zeta(I, J, t)\}$
<small>235</small> which satisfy

$$\frac{d\mathbf{O}(\mathbf{R}, t)}{dt} = F_{\mathbf{R}}(\mathbf{S}(\mathbf{r}, t), \theta) + [\mathcal{F}(\mathbf{R}, t), 0],$$
$$\frac{d\mathbf{O}(I, J, t)}{dt} = F_{I,J}(\mathbf{S}(i, j, t), \theta) + [\mathcal{F}(I, J, t), 0] \qquad (4)$$

<small>236</small> The number of regional observed variables is $D_R = 3(N_x \times N_y) \leq D_G$.
<small>237</small> $F_{\mathbf{R}}(\mathbf{S}(\mathbf{r}, t), \theta) = F_{I,J}(\mathbf{S}, \theta)$ is the (global) vector field restricted to the region
<small>238</small> $\mathbf{R}$. It is a function of the states of the global dynamics $\mathbf{S}(i, j, t)$.

<small>239</small>    Using data from observations on the $\mathbf{O}(\mathbf{R}, t)$, without knowledge of the
<small>240</small> vector field $F_{I,J}(\mathbf{S}(i, j, t), \theta)$, we want to construct a discrete time dynamical
<small>241</small> rule which takes $\mathbf{O}(\mathbf{R}, t)$ forward in time; $\mathbf{O}(\mathbf{R}, t) \rightarrow \mathbf{O}(\mathbf{R}, t + \Delta t) = \mathbf{O}(\mathbf{R}, t + $
<small>242</small> $h)$. This discrete time dynamical map is our forecasting system for the region.

<small>243</small>    Observations are made at $N_O$ times $t_n = t_0 + nh; n = 0, 1, ..., N_O - 1$
<small>244</small> giving us $\mathbf{O}(\mathbf{R}, t_n) = \mathbf{O}(\mathbf{R}, n)$ at all those times. These observations form a
<small>245</small> trajectory in $D_R \leq D_G$ dimensional space. We are interested in the situation
<small>246</small> where $D_R < D_G$, giving us observations in a subregion of the global dynam-
<small>247</small> ics. For purposes of explaining the steps in the construction of a regional
<small>248</small> forecasting system we will first consider the case $D_R = D_G$. A return to
<small>249</small> $D_R < D_G$ will follow that discussion.

<small>250</small>    Now we integrate the regional dynamical equation Eq. (4) over the in-
<small>251</small> terval $[t_n, t_n + h] = [t_n, t_{n+1}]$. This gives us the *flow* of the $D_R$ dimensional
<small>252</small> dynamical system which we find to be

$$\mathbf{O}(\mathbf{R}, t_n + h) = \mathbf{O}(\mathbf{R}, n + 1) = \mathbf{O}(\mathbf{R}, n)$$
$$+ \int_{t_n}^{t_n + h} dt' \left\{ F_{\mathbf{R}}(\mathbf{S}(\mathbf{r}, t'), \theta) + [\mathcal{F}(\mathbf{R}, t'), 0] \right\}, \qquad (5)$$

<small>253</small>    As we do not know the vector field $F_{I,J}(\mathbf{S}(\mathbf{r}, t), \theta)$, we must **represent**
<small>254</small> the integral over it in some manner. The integral over the external forces on
<small>255</small> the fluid in the subregion, $\mathbf{R}$, $[\mathcal{F}(\mathbf{R}, t), 0]$ we can approximate, because, to
<small>256</small> solve the original global problem, we were required to specify how the fluid
<small>257</small> was driven both regionally and globally Roeckner et al. (2003); Staff (2021).
<small>258</small> The forcing of the fluid is an aspect of the flow that is not intrinsic to the

<small>10</small>



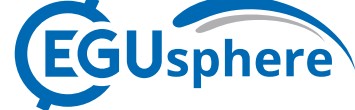

fluid properties themselves, and it is important to observe that this external forcing is additive. Essentially these are just Newton's equations of motion.

Using whatever approximation to the integral over $\mathbf{F}_{I,J}(\mathbf{S}(\mathbf{r},t'),\theta)$ one wishes Olver (2017) that establishes it as dependent on $\mathbf{S}(\mathbf{r},t)$, so the resulting dynamical rule for moving forward in time is explicit, we arrive at a representation of the flow as the discrete time map $(1 \leq i \leq n_x, 1 \leq j \leq n_y :$ $1 \leq I \leq N_x \leq n_x, 1 \leq J \leq N_y \leq n_y.)$

$$\mathbf{O}(I, J, n+1) = \mathbf{O}(I, J, n) + \mathbf{f}_{I,J}(\mathbf{S}(i, j, , n), \boldsymbol{\chi})$$
$$+\frac{h}{2}\bigg[[\mathcal{F}(I, J, n), 0] + [\mathcal{F}(I, J, n+1), 0]\bigg]. \tag{6}$$

Although we do not use it in our construction of DDF models, we could have chosen Equation (6) to have $\mathbf{S}(i, j, n+1)$ on the right hand side as we represent the integral over the vector field $\mathbf{F}_{I,J}(\mathbf{S}(i, j, t)$ in Equation (5) Olver (2017). The dynamical discrete time map would them be *implicit* and using it would require the solution of a nonlinear problem at each step.

The trapezoidal rule was used to approximate the integral over the known forces which should be quite adequate as the measurements are known only at intervals of size h. The $\boldsymbol{\chi}$ are constants to be estimated in the representation of the flow of the discrete time dynamics. We call $\mathbf{f}_{I,J}(\mathbf{S}(i, j, n), \boldsymbol{\chi})$ the vector field of the discrete time flow restricted to the regional variables.

## 2.3 Proceeding in Two Steps

As noted earlier, we proceed now in two steps:

1. First we select $D_R = D_G$ putting us in the situation where we observe all components of the global dynamics, but in which we do not know the equations of these global dynamics but only have the data.

2. Second we move to the case of interest in this paper when $D_R < D_G$ where we are also required to find a way to introduce information about the global dynamical flow into our rule $\mathbf{O}(\mathbf{R}, t) \rightarrow \mathbf{O}(\mathbf{R}, t + \Delta t)$, Equation (6).

### 2.3.1 First Step, $D_R = D_G$; Introducing Radial Basis Functions

This is the case where we observe all of the global set of state variables $\mathbf{S}(\mathbf{r}, n)$. This is not of central interest to us, but it is nonetheless very instructive for



| Parameter | Symbol | Value |
|---|---|---|
| Fluid Density | $\rho$ | 1 kg/m$^3$ |
| Coriolis Parameter | $f_0 + \beta y$ | $f_0 = 10^{-5} 1/s$ |
| | | $\beta = 10^{-12} 1/m\,s$ |
| Effective Viscosity | A | 100 m$^2$/s |
| Rayleigh friction | $\epsilon$ | $10^{-8}$ 1/s |
| Gravity | g | 9.8 m/s$^2$ |
| Time Step | h | 6 min |
| Resting Fluid Depth | $H_0$ | 50 m |
| Grid Resolution | $\Delta x = \Delta y$ | 100 km |
| Domain Extent | $L_x = n_x \Delta x; L_y = n_y \Delta y$ | |
| Body Forcing | $[\mathcal{F}_1(\mathbf{r}, t), \mathcal{F}_2(\mathbf{r}, t)]$ | $[F_1(\mathbf{r}, t), 0]\ m/s^2$ |
| | $F_1(\mathbf{r}, t) = -F_0 \cos(\frac{2\pi y}{L_y})$ | $F_0 = 10^{-5} m/s^2$ |
| Number of Centers | $N_c$ | 1000 |

Table 1: Table of Parameters and Symbols in the shallow water flow. For the RBF representation we have additional parameters $\boldsymbol{\chi}$ used in the solution to the DDF flow vector field. These depend on the particular problem, e.g. which subregion one selects. The $\boldsymbol{\chi}$ include the RBF Shape Factor, $\sigma$, the time Time Delay $\tau = T_h h; (T_h = \text{integer})$, the Embedding Dimension, $D_E$, and the Tikhonov Regularization Parameter, B. We have used $N_c = 1000$.

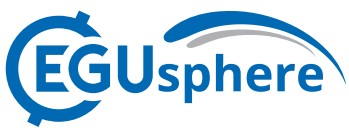

<sup></sup>288 making our next step.

<sup></sup>289 The discrete time dynamics appears in this instance as

$$\mathbf{S}(i,j,n+1) = \mathbf{S}(i,j,n) + \mathbf{f}_{i,j}(\mathbf{S}(r,n),\boldsymbol{\chi})$$
$$+\frac{h}{2}\Big[[\mathcal{F}(\mathbf{r},n),0] + [\mathcal{F}(\mathbf{r},n+1),0]\Big], \tag{7}$$

<sup></sup>290 and we need a representation of the discrete time vector field $\mathbf{f}_{i,j}(\mathbf{S}(i,j,n),\boldsymbol{\chi})$
<sup></sup>291 of the flow.

<sup></sup>292 This flow vector field $\mathbf{f}(\mathbf{S},\boldsymbol{\chi})$ is a set of $D_G$ functions on the $\mathbf{S} \in \mathcal{R}^{D_G}$.
<sup></sup>293 There are many ways to represent such functions of many variables. We can
<sup></sup>294 think of the observed samples as points of information about a distribution
<sup></sup>295 $\mathbf{f}(\mathbf{S},\boldsymbol{\chi})$ and ask that the representation give us an interpolating function
<sup></sup>296 among the observed point locations $\mathbf{S}(\mathbf{r},t_n) = \mathbf{S}(\mathbf{r},n)$.

<sup></sup>297 We select a method of representing this function using radial basis func-
<sup></sup>298 tions (RBFs) Hardy (1971); Micchelli (1986); Broomhead and Lowe (1988);
<sup></sup>299 Schaback (1995); M. D. J. (2002); Buhmann (2009). In using this method
<sup></sup>300 we use a K-means clustering algorithm Du and Swamy (2006) to select $N_c$ of
<sup></sup>301 all the $N_O$ observed locations in $\mathbf{S}$ space. These are called centers. The flow
<sup></sup>302 vector field is then given in components by

$$f_a(\mathbf{S},\boldsymbol{\chi}) = \sum_{k=1}^{K} c_{ak}p_k(\mathbf{S}) + \sum_{q=1}^{N_c} w_{aq}\boldsymbol{\psi}((\mathbf{S}-\mathbf{S}(q))^2,\sigma), a = 1,2,...,D_G \tag{8}$$

<sup></sup>303 In this $p_k(\mathbf{S})$ is a polynomial of degree k. The radial basis functions
<sup></sup>304 $\boldsymbol{\psi}((\mathbf{S}-\mathbf{S}(q))^2,\sigma)$ are sensitive to the distribution samples, the centers $\mathbf{S}(q)$,
<sup></sup>305 over a range $\sigma$.

<sup></sup>306 There are a multitude of choices for the RBF $\boldsymbol{\psi}((\mathbf{S}-\mathbf{S}(q))^2,\sigma)$, and we
<sup></sup>307 have investigated the use of two. The Gaussian

$$\boldsymbol{\psi}_G((\mathbf{S}-\mathbf{S}(q))^2,\sigma) = \exp[-R(\mathbf{S}-\mathbf{S}(q))^2]; \ R = \frac{1}{2\sigma^2}, \tag{9}$$

<sup></sup>308 and the multiquadric of Hardy Hardy (1971)

$$\boldsymbol{\psi}_{MQ}((\mathbf{S}-\mathbf{S}(q))^2,\sigma) = \sqrt{(\mathbf{S}-\mathbf{S}(q))^2 + \sigma^2}. \tag{10}$$

<sup></sup>309 The constants $\boldsymbol{\chi} = \{c_{ak}, w_{aq}\}$ are determined by the linear algebra prob-





lem

$$\mathbf{S}(n+1) = \mathbf{S}(n) + \sum_{k=1}^{K} c_{ak} p_k(\mathbf{S}(n)) + \sum_{q=1}^{N_c} w_{aq} \boldsymbol{\psi}((\mathbf{S}(n) - \mathbf{S}(q))^2, \sigma)$$

$$+ \frac{h}{2}\bigg[ [\mathcal{F}(\mathbf{r}, n), 0] + [\mathcal{F}(\mathbf{r}, n+1), 0] \bigg], \tag{11}$$

for $n = 0, 1, 2, ..., N_O - 1$ and $a = 1, 2, ..., D_G$.

### 312  2.3.2  Second Step, $D_R < D_G$; Utilizing Time Delay Embedding

Turning back to the question of developing a dynamical map for the regional
subset $\mathbf{O}(\mathbf{R}, n)$ of our dynamical variables, we see that the observations are a
projection from dimension $D_G \rightarrow D_R < D_G$. To proceed we require a space
of state variables equivalent to the full state space of the $\mathbf{S}(\mathbf{r}, t)$, so we must
'unproject' the $\mathbf{O}(\mathbf{R}, t)$ to a space equivalent to $\mathbf{S}(\mathbf{r}, t)$.

A dynamical method for accomplishing this is well analyzed in the non-
linear dynamics literature. It rests on the fact that as the observed quantities
move from some time $t - \tau$ to time t, they depend on all of the state variables
$\mathbf{S}(\mathbf{r}, t)$ as seen in Equation (6). Using time delays of the observed regional
variables provides us the desired information on the unobserved state vari-
ables.

This suggests creating a $D_E$-dimensional time delay embedding space Tak-
ens (1981); Eckmann and Ruelle (1985); Sauer et al. (1991); Abarbanel
(1996); Kantz and Schreiber (2004) with vectors of dimension $D_{TD} = D_R D_E$

$$\mathbf{TD}(t) = [\mathbf{O}(\mathbf{R}, t), \mathbf{O}(\mathbf{R}, t - \tau), \mathbf{O}(\mathbf{R}, t - 2\tau), ..., \mathbf{O}(\mathbf{R}, t - (D_E - 1)\tau]. \tag{12}$$

This vector of time delays depends only on the observed quantities in the
region labeled by $\mathbf{R}$ and their time delays. It is through those time delays
that $\mathbf{TD}$ inherits information about the dynamics outside the region $\mathbf{R}$.

Using this time delay vector in the observed dynamics gives us

$$\mathbf{O}(\mathbf{R}, n+1) = \mathbf{O}(\mathbf{R}, n) +$$

$$\mathbf{f}_{\mathbf{R}}(\mathbf{TD}(n), \boldsymbol{\chi}) + \frac{h}{2}\bigg[ [\mathcal{F}(\mathbf{R}, n), 0] + [\mathcal{F}(\mathbf{R}, n+1), 0] \bigg]. \tag{13}$$

It is useful to write this in components. The $\mathbf{O}(\mathbf{R}, t)$ are $D_R$ dimensional
$\mathbf{O}(\mathbf{R}, t) = \{O_\alpha(t)\}$; $\alpha = 1, 2, ..., D_R$. The dynamical map for the regional





observables becomes

$$O_\alpha(n+1) = O_\alpha(n) +$$
$$f_\alpha(\mathbf{TD}(n), \boldsymbol{\chi}) + \frac{h}{2}\left[[\mathcal{F}(\mathbf{R}, n), 0] + [\mathcal{F}(\mathbf{R}, n+1), 0]\right]_\alpha. \quad (14)$$

This informs us that we will need a representation for each of the $D_R$
components of the observed regional variables. The DDF parameters we need
to estimate using the observed data now include $\boldsymbol{\chi} = \{w_{\alpha q}, c_{\alpha j}, \sigma, B, D_E, T_h\}$.

## 3   Estimating $\boldsymbol{\chi}$ by Minimizing C($\boldsymbol{\chi}$)

The $\mathbf{O}(\mathbf{R}, t)$ are $D_R$ dimensional $\mathbf{O}(\mathbf{R}, t) = \{O_\alpha(t)\};\ \alpha = 1, 2, ..., D_R$. The
dynamical map for the regional observables is

$$O_\alpha(n+1) = O_\alpha(n) +$$
$$f_\alpha(\mathbf{TD}(n), \boldsymbol{\chi}) + \frac{h}{2}\left[[\mathcal{F}(\mathbf{R}, n), 0] + [\mathcal{F}(\mathbf{R}, n+1), 0]\right]_\alpha$$
$$\mathbf{TD}(t) = [\mathbf{O}(\mathbf{R}, t), \mathbf{O}(\mathbf{R}, t-\tau), ..., \mathbf{O}(\mathbf{R}, t-(D_E-1)\tau)]. \quad (15)$$

To estimate the parameters $\boldsymbol{\chi}$ in the DDF forecasting function, we form
the cost function $C(\boldsymbol{\chi})$ and minimize this objective function with respect to
the elements of $\boldsymbol{\chi} = \{w_{\alpha q}, c_{\alpha j}, R, B, \tau = hT_h, D_E\}$.

$$C(\boldsymbol{\chi}) = \sum_{N_c+1}^{N_O} \Bigg\{ [\mathbf{O}(n+1) - \mathbf{O}(n) - \mathbf{f}(\mathbf{TD}(n), \boldsymbol{\chi})$$
$$-\frac{h}{2}\bigg[[\mathcal{F}(\mathbf{R}, n), 0] + [\mathcal{F}(\mathbf{R}, n+1), 0]\bigg]\Bigg\}^2. \quad (16)$$

In Equation (16)

$$f_\alpha(\mathbf{TD}(n), \boldsymbol{\chi}) = \sum_{q=1}^{N_c} w_{\alpha q}\boldsymbol{\psi}((\mathbf{TD}(n) - \mathbf{TD}^c(q))^2, \sigma) + \sum_{j=1}^{D_R} c_{\alpha j}\mathbf{O}_j(n) \quad (17)$$

Even though we can choose any RBF for $\boldsymbol{\psi}$, the $C(\boldsymbol{\chi})$ is always linear in
the weights $w_{\alpha q}, c_{\alpha j}$, enabling the use of the linear algebra of Ridge Regression
or Tikhonov regularization in estimating them.





The other elements of $\boldsymbol{\chi}$ enter the minimization of $C(\boldsymbol{\chi})$ nonlinearly.
There are many excellent ways to search for values for these using ideas noted
in the Appendix. We utilized a rather coarse grained grid search to identify
good forecasts, then refined the search in those regions. This works well, as
the results we present below demonstrate. Nonetheless, we recommend more
efficient methods. Again, see the Appendix.



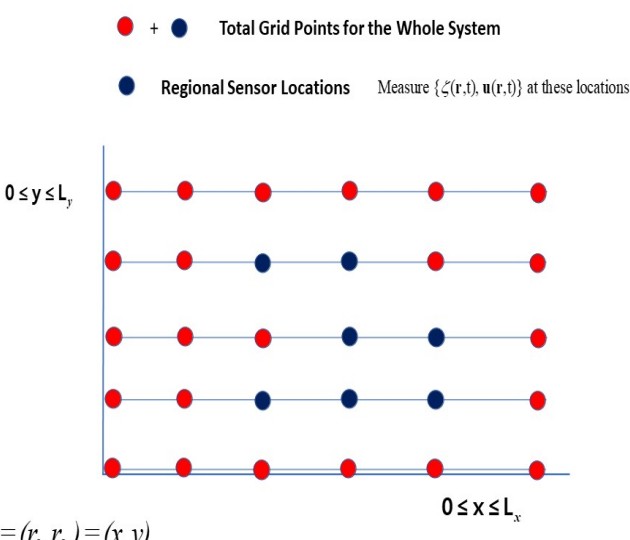

Figure 1: Overall Scheme of Regional DDF Forecasting in the context of the SWE. Red dots are grid points of the overall dynamical system, Eq. (3). Blue dots are the location of the "regional sensors" where fluid height $\zeta(\mathbf{r}, t)$ and fluid velocity $\mathbf{v}(\mathbf{r}, t) = \{v_1(\mathbf{r}, t), v_2(\mathbf{r}, t)\}$ are recorded. If there are $N_R$ regional sensor points then there are $D_R = 3N_R$ regional measured time series. The spatial locations for the sensors are denoted $\mathbf{R}$; these are the blue locations on the grid. In this Figure $n_x \times n_y = 30$; $N_R = 7$. In this example there would then be $D_R = 3N_R = 21$ observed regional time series out of $D_G = 3(n_x \times n_y) = 90$ global time series. $D_R < D_G$.





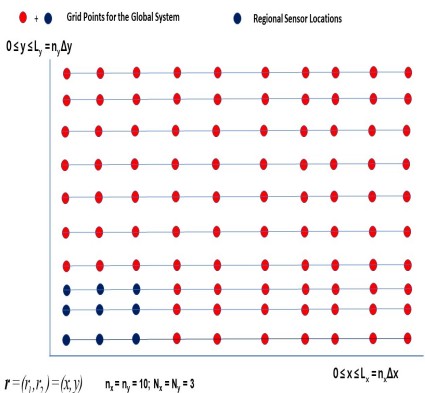

Figure 2: Shallow Water Equations on a $(n_x = 10) \times (n_y = 10)$ grid. $D_G = 3(n_x \times n_y)$. Red dots are grid points of the overall global dynamical system, Eq. (3), a set of Shallow Water Equations on a $\beta$ plane. Blue dots are the location of the "regional sensors" where fluid height $\zeta(\mathbf{r}, t)$ and fluid velocity $\mathbf{v}(\mathbf{r}, t)$ are recorded. There are $(N_x = 3) \times (N_y = 3) = 9$ regional sensor locations in this example, and $D_R = 3(N_x \times N_y) = 27$ measured time series. The spatial locations of the sensors are denoted $\mathbf{R}$; these are the blue locations on the grid. In this Figure $n_x \times n_y = 100$, and $D_G = 3(n_x \times n_y) = 300$. We have 27 observed regional time series out of 300 global time series. $D_R < D_G$.

# 4 Results from the Example of the SWE on a $\beta$ Plane

## 4.1 Clustered Sensor Region; 3x3 Corner

The shallow water equations were solved using the method of Sadourny Sadourny (1975) with forcing, viscous dissipation, Coriolis forces, and Rayleigh friction as indicated in Equation (1). We used periodic boundary conditions.

The global grid is $n_x = n_y = 10$, so we solved 300 ODEs to generate the time series for the state variables $\mathbf{S}(\mathbf{r}, t)$ on this grid. A regional sub grid with $N_x = N_y = 3$ was selected, and on this regional subgrid we 'measured' $\{u(\mathbf{R}, t), v(\mathbf{R}, t), \zeta(\mathbf{R}, t)\}$ to form the $D_R = 27$ dimensional observation vectors $\mathbf{O}(\mathbf{R}, t)$. The regional sensor locations are shown in Figure (2). We



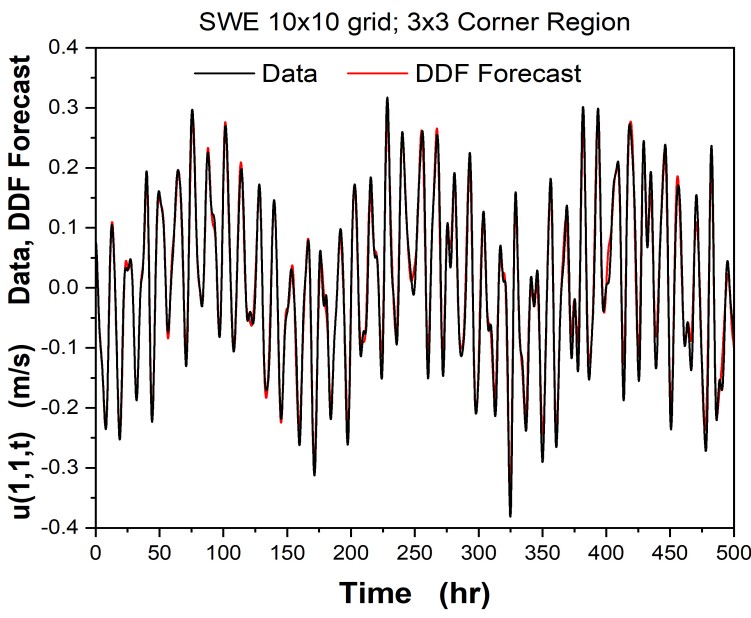

Figure 3: SWE on a $10 \times 10$ grid. The sensor region is comprised of 9 locations, blue dots in Figure (2), in a 3x3 corner location. The sensor region is comprised of 9 locations, blue dots in Figure (8). We display the data and the DDF forecast for the x-velocity $u(1,1,t)$. The time delay parameters here are $D_E = 20, \tau = 20\Delta t$ with $\Delta t = h = 0.1 hr$.

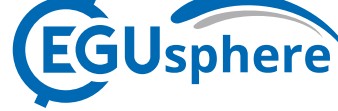



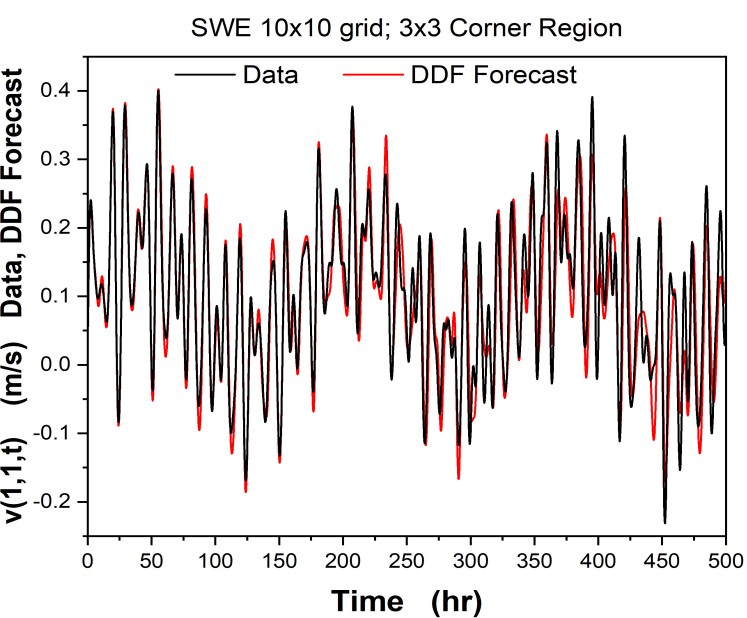

Figure 4: SWE on a $10 \times 10$ grid. The sensor region is comprised of 9 locations, blue dots in Figure (2), in a 3x3 corner location. We display the data and the DDF forecast for the y-velocity $v(1, 1, t)$. The time delay parameters here are $D_E = 20, \tau = 20\Delta t$ with $\Delta t = h = 0.1 hr$.



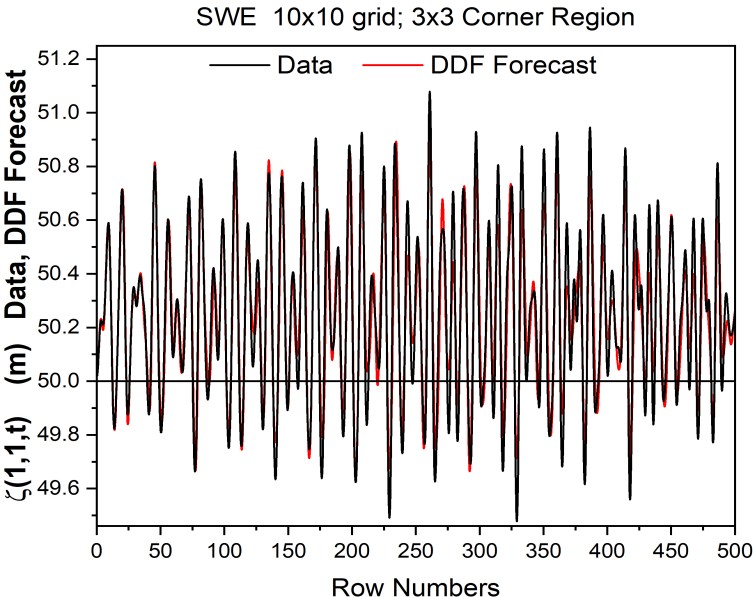

Figure 5: SWE on a $10 \times 10$ grid. The sensor region is comprised of 9 locations, blue dots in Figure (2), in a 3x3 corner location. We display the data and the DDF forecast for the fluid height $\zeta(1, 1, t)$. The fluid rest height is $H_0 = 50$m. The time delay parameters here are $D_E = 20, \tau = 20\Delta t$ with $\Delta t = 0.1hr$.





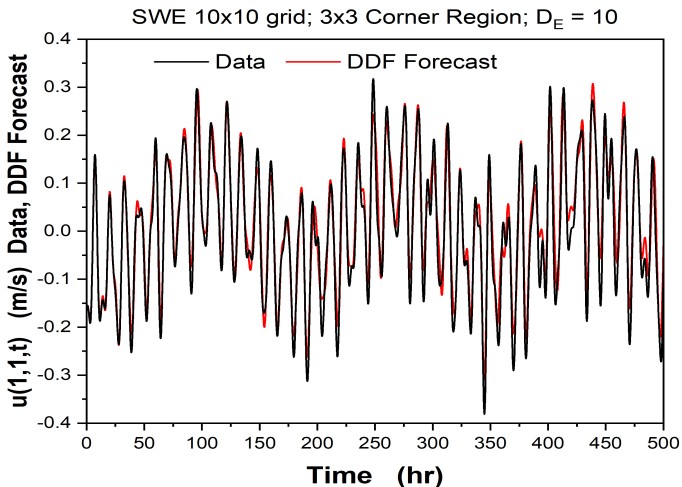

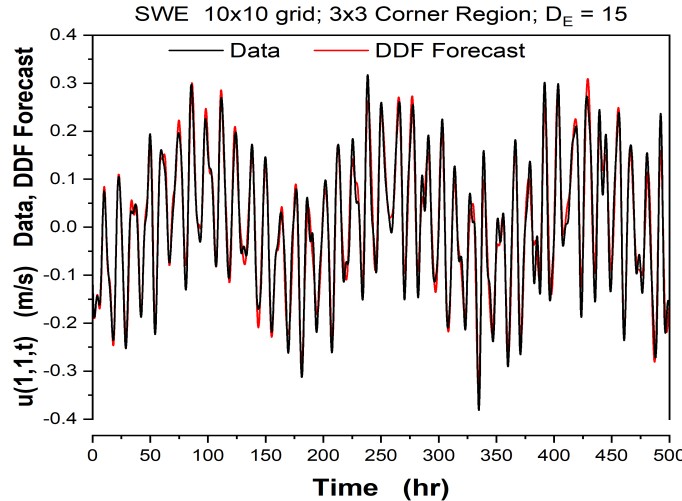

Figure 6: SWE on a $10 \times 10$ grid. The sensor region is comprised of 9 locations, blue dots in Figure (2), in a 3x3 corner location. We display the data and the DDF forecast for the x-velocity $u(1, 1, t)$. $\tau = 20\Delta t$ with $\Delta t = h = 0.1 hr$. **Top Panel** $D_E = 10$ **Bottom Panel** $D_E = 15$. The results with $D_E = 20$ are in Figure (3).



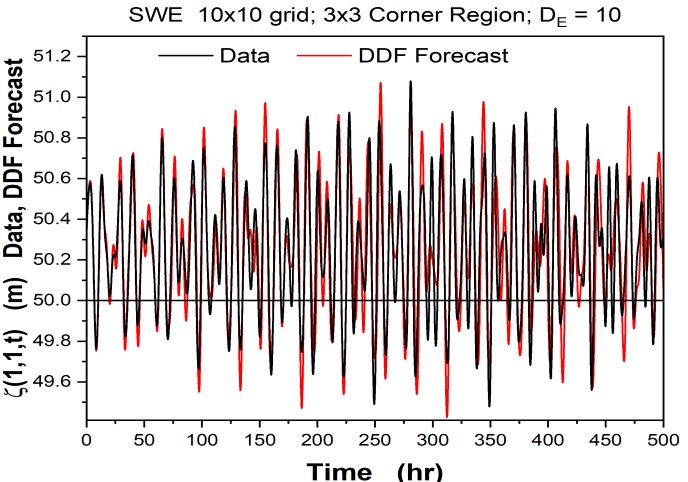

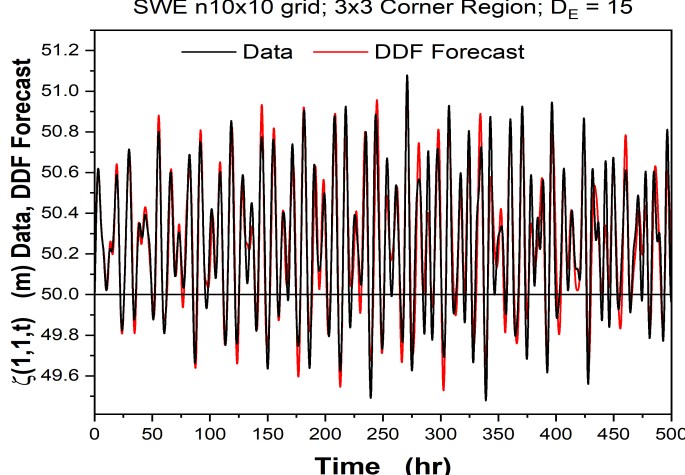

Figure 7: SWE on a $10 \times 10$ grid. The sensor region is comprised of 9 locations, blue dots in Figure (2), in a 3x3 corner location. We display the data and the DDF forecast for the fluid height $\zeta(1, 1, t)$. The fluid rest height is $H_0 = 50$m. $\tau = 20\Delta t$ with $\Delta t = h = 0.1 hr$. **Top Panel** $D_E = 10$ **Bottom Panel** $D_E = 15$. The results with $D_E = 20$ are in Figure (5).



generated $N_O = 15,000$ time steps of size 0.1 hr. $N_c = 1000$ centers were se-
lected from these data.These were used in a Polynomial plus Gaussian RBF,
Equation (8). 1000 hr of these data were used to train the RBF representa-
tion of the discrete time flow vector field, the 500 hr of forecasts were made
of the 9 regional state variables.

In the estimation of the parameters we found B = 1.0 x10$^{-9}$, R = 1.0 x
10$^{-6}$, so $\sigma \approx 707$, $D_E = 20$ and $\tau = 20h = 2hr$ gave the best forecasts. See
Figure (3), Figure (4), and Figure (5) for the data and the DDF Forecast for
$u(1,1,t), v(1,1,t),$ and $\zeta(1,1,t)$ with $D_E = 20$.

We note that values of the embedding dimension $10 \leq D_E \leq 20$ gave
forecasts of more or less equal quality. In Figure (6) and then Figure (7) we
display the x-velocity $u(1,1,t)$ for $D_E = 10$ and $D_E = 15$ respectively, and
then $\zeta(1,1,t)$ for $D_E = 10$ and $D_E = 15$ respectively.




| State Variable | $D_E$ | RMS Error |
|---|---|---|
| u(1,1,t) | 10 | 0.034 m/s |
| u(1,1,t) | 15 | 0.028 m/s |
| u(1,1,t) | 20 | 0.02 m/s |
| $\zeta(1,1,t)$ | 10 | 0.37 m |
| $\zeta(1,1,t)$ | 15 | 0.37 m |
| $\zeta(1,1,t)$ | 20 | 0.32 m |

Table 2: 3x3 Corner example, Figure (2). RMS error in DDF forecasting of
the state variable $u(1,1,t)$ and $\zeta(1,1,t)$ as we vary $D_E$. These are evaluated
in the prediction region; N = 5000 h = 500 hr.

The RMS error is evaluated as

$$RMS(u) = \sqrt{\frac{1}{N}\sum_{n=1}^{N}\left[u_{Data}(1,1,n) - u_{DDF}(1,1,n)\right]^2} \; m/s,$$

$$RMS(\zeta) = \sqrt{\frac{1}{N}\sum_{n=1}^{N}\left[\zeta_{Data}(1,1,n) - \zeta_{DDF}(1,1,n)\right]^2} \; m, \qquad (18)$$

and the results are in Table 2.

In the underlying mathematical theorem associated with time delay em-
bedding Takens (1981); Eckmann and Ruelle (1985); Sauer et al. (1991);
Abarbanel (1996); Kantz and Schreiber (2004) there is no restriction on the
time delay $\tau$. From a geometrical result one knows that if the dimension
of $\mathbf{TD}(n)$, namely $D_E$ is large enough, then the unprojection we wish to
achieve is accomplished. The results of Sauer et al. (1991) indicate that if
the dimension is larger than $2D_A + 1$, with $D_A$ the information dimension of
the strange attractor, the unprojection will work. We do not know the $D_A$
of the global SWE model, but if it is near 300 or so, then $D_E \approx 10 - 20$ is a
consistent with this.

In the next Figures, we compare the DDF regional forecasts with the
'data' comprised of the solutions to the 300 SWE ODEs at the regional grid
points. In this first set of results, we choose the sensor region to be comprised
of small clusters of sites.

In Figure (3) we display the x-velocity $u(1,1,t)$ at the regional grid loca-
tion (1,1) as forecast by the DDF dynamics trained as indicated. In Figure





<sub>395</sub> (4) we make the same comparison for the y-velocity $v(1, 1, t)$ at the (1,1)
<sub>396</sub> regional grid location. In Figure (5) the same comparison is made for the
<sub>397</sub> height of the fluid $\zeta(1, 1, t)$. The height of the fluid when at rest is $H_0 = 50m$.

## <sub>398</sub> 4.2 Region With Sparse, Distributed Sensors

<sub>399</sub> In the next set of outcomes for DDF regional forecasting, we now select our
<sub>400</sub> sensors to be disperesed over the 'global' region. Figure (8) shows ten sensors
<sub>401</sub> dispered among the 100 grid locations of the global dynamical regime. The
<sub>402</sub> senor locations were selected at random among the 100 possible sites available
<sub>403</sub> on the 'global' grid. The main purpose of this example is to demonstrate that
<sub>404</sub> the sensor sites where $\{v_1(\mathbf{R}, t), v_2(\mathbf{R}, t), \zeta(\mathbf{R}, t)\}$ are observed, in the SWE
<sub>405</sub> example, need not all be contiguous.

<sub>406</sub>   The sucess of DDF forecasting using sensors in broadly dispered sensor
<sub>407</sub> region $\mathbf{R}$ suggests one could use the strategy to forecast in a quite broad
<sub>408</sub> geographical sub-region of a global dynamical system.

## <sub>409</sub> 4.3 Off Center; 2x2 Region

<sub>410</sub> Our final example is a 2x2 subregion randomly selected to be off center in
<sub>411</sub> the global 10x10 grid. This is shwon in Figure (12).





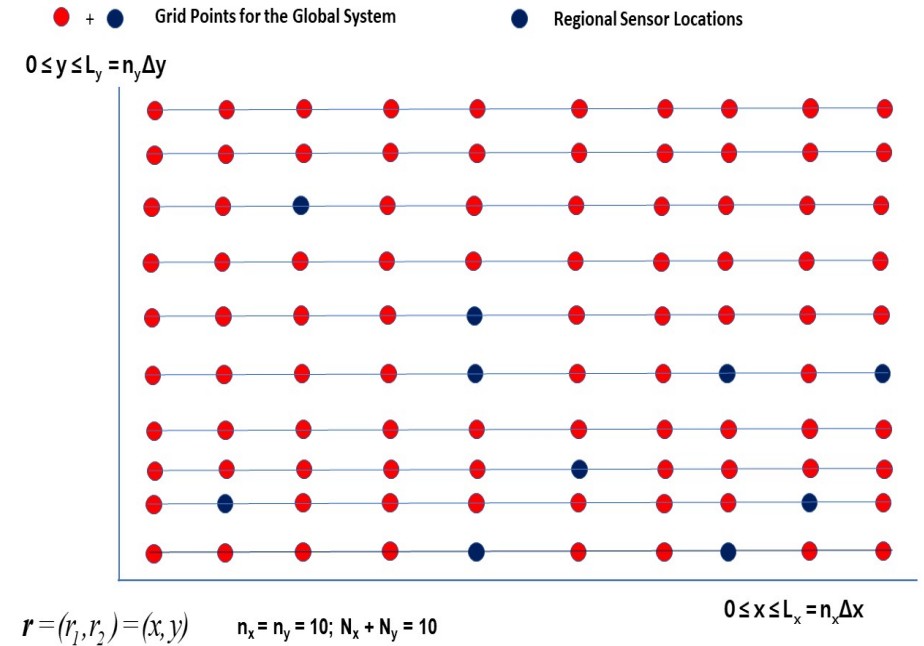

Figure 8: Sparse Regional Sensor Locations. SWE on a $10 \times 10$ grid. The sensor region, indicated by blue dots, is comprised of 10 locations selected at random among the 100 global grid points.



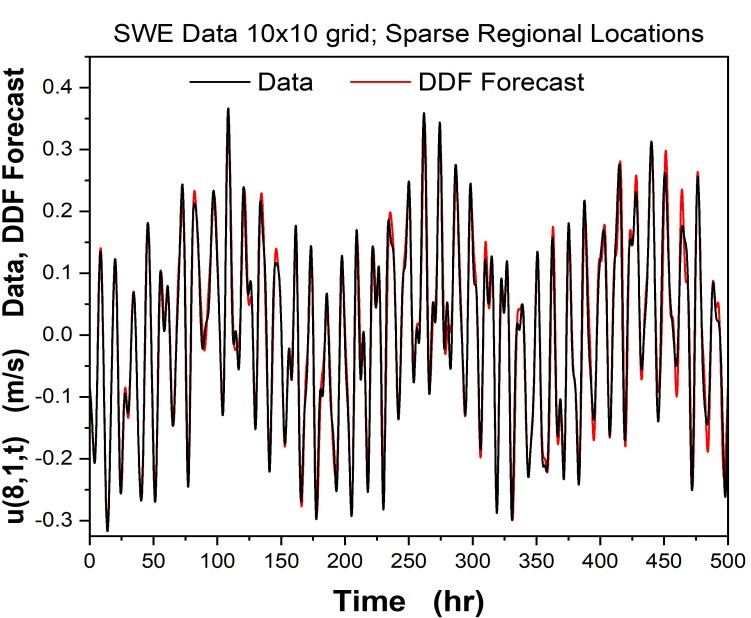

Figure 9: SWE on a $10 \times 10$ grid. The sensor region is comprised of 10 locations, blue dots in Figure (8), selected at random among the 100 global grid points. We display the data and the DDF forecast for the x-velocity $u(8, 1, t)$. The time delay parameters here are $D_E = 20, \tau = 17h$; $h = 0.1hr$.



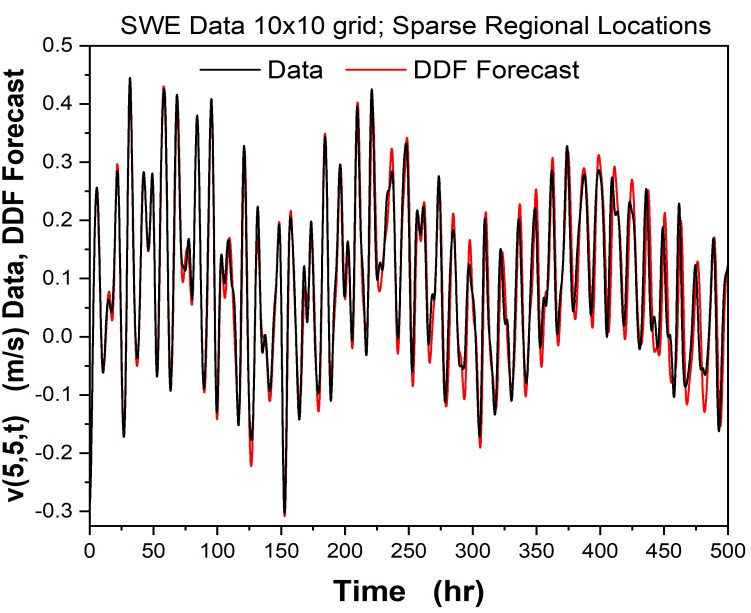

Figure 10: SWE on a $10 \times 10$ grid. The sensor region is comprised of 10 locations, blue dots in Figure (8), selected at random among the 100 global grid points. We display the data and the DDF forecast for the y-velocity $v(5, 5, t)$. The time delay parameters here are $D_E = 20, \tau = 17\Delta t$ with $\Delta t = 0.1hr$.



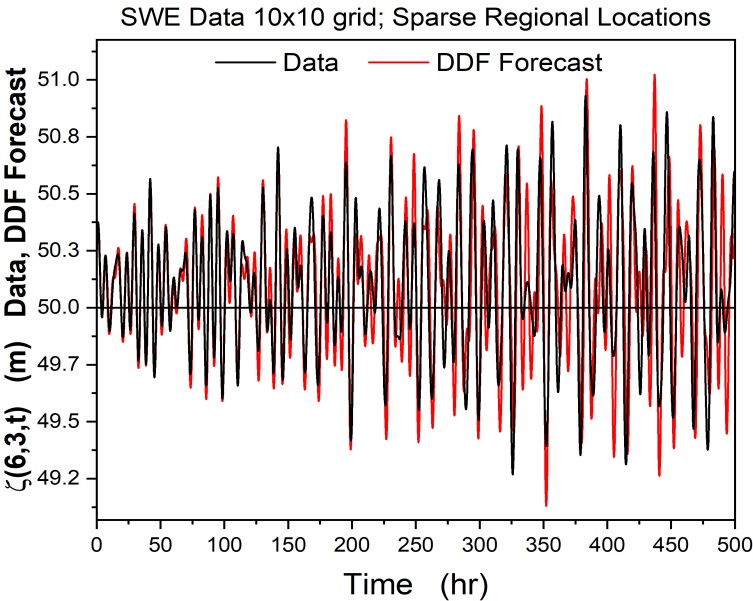

Figure 11: SWE on a $10 \times 10$ grid. The sensor region is comprised of 10 locations, blue dots in Figure (8), selected at random among the 100 global grid points. We display the data and the DDF forecast for the fluid height $\zeta(6, 3, t)$. The fluid rest height is $H_0 = 50$m. The time delay parameters here are $D_E = 20, \tau = 17\Delta t$ with $\Delta t = 0.1hr$.





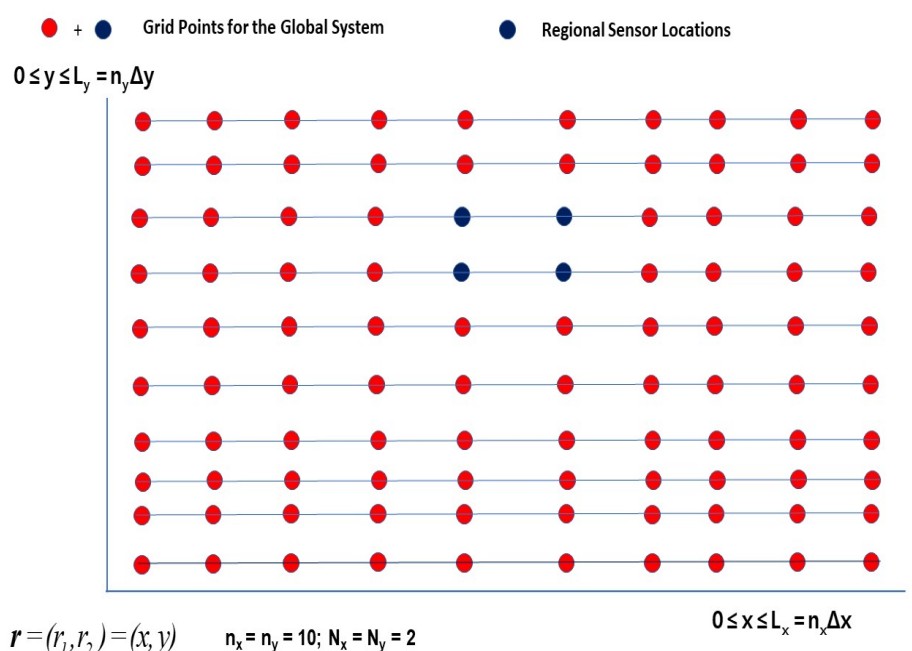

Figure 12: SWE global 10x10 grid. Regional grid is 2x2 and is located off center within the global grid.

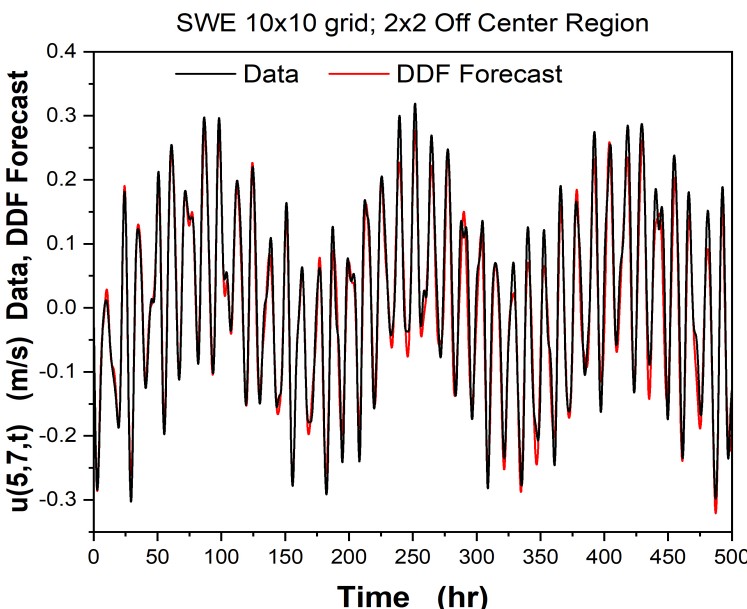

Figure 13: SWE on a $10 \times 10$ grid. The sensor region is comprised of 4 locations, blue dots in Figure (8), in a 2x2 off center location. The sensor region is comprised of 4 locations, blue dots in Figure (8), in a 2x2 off center location. We display the data and the DDF forecast for the x-velocity $u(5, 7, t)$. The time delay parameters here are $D_E = 20, \tau = 17\Delta t$ with $\Delta t = 0.1 hr$.

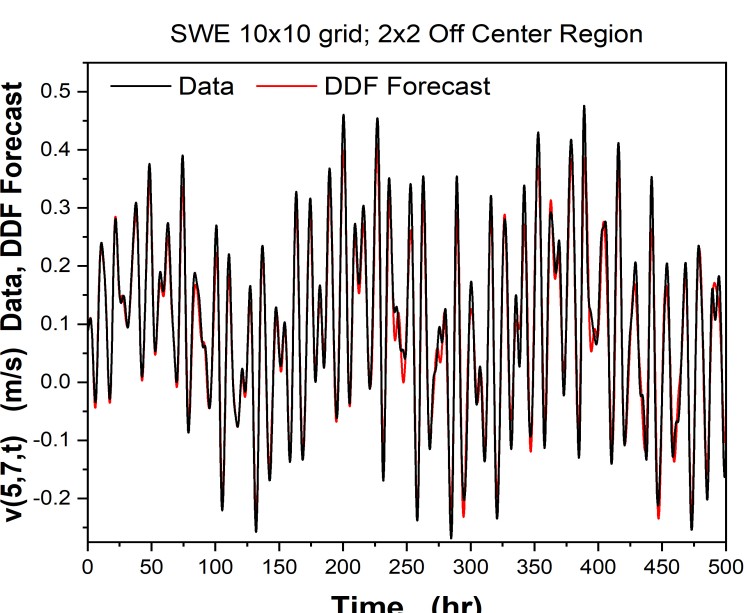

Figure 14: SWE on a $10 \times 10$ grid. The sensor region is comprised of 4 locations, blue dots in Figure (8), in a 2x2 off center location. We display the data and the DDF forecast for the y-velocity $v(5, 7, t)$. The time delay parameters here are $D_E = 20, \tau = 17\Delta t$ with $\Delta t = 0.1hr$.

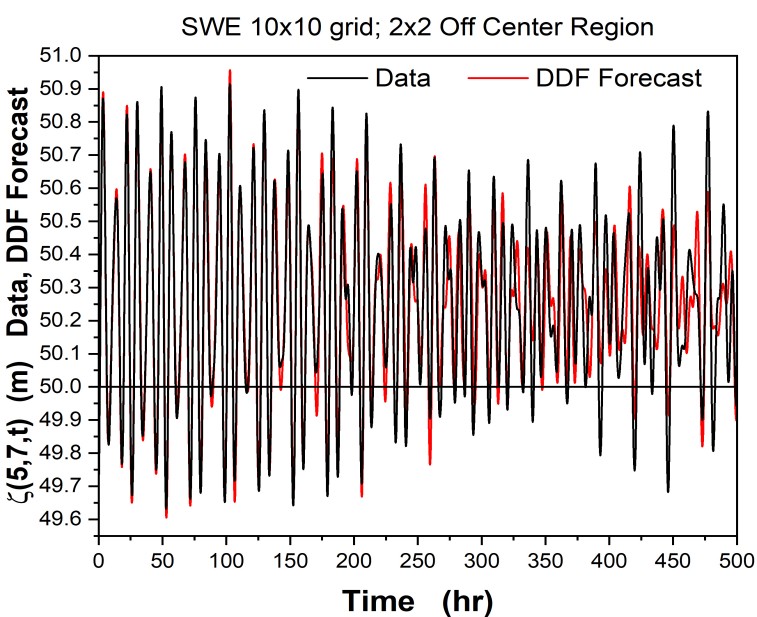

Figure 15: SWE on a $10 \times 10$ grid. The sensor region is comprised of 4 locations, blue dots in Figure (8), in a 2x2 off center location. We display the data and the DDF forecast for the fluid height $\zeta(5, 7, t)$. The fluid rest height is $H_0 = 50\text{m}$. The time delay parameters here are $D_E = 20, \tau = 17\Delta t$ with $\Delta t = 0.1 hr$.

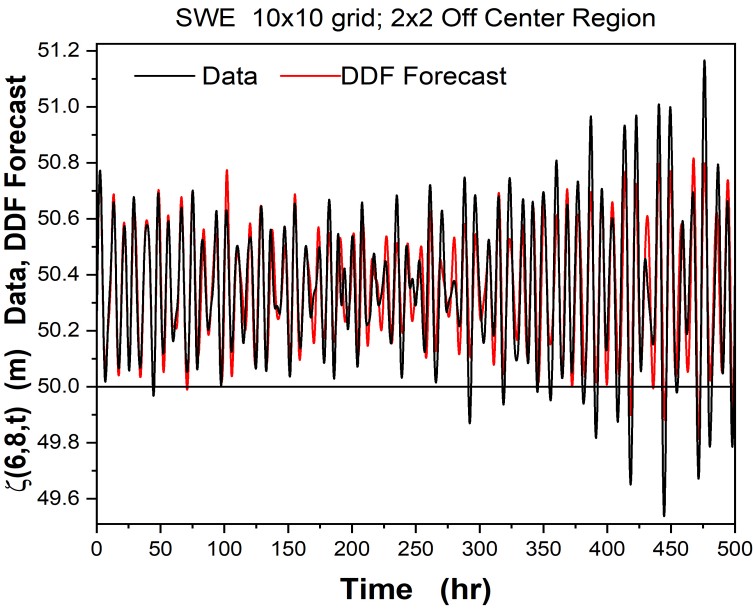

Figure 16: SWE on a $10 \times 10$ grid. The sensor region is comprised of 4 locations, blue dots in Figure (8), in a 2x2 off center location. We display the data and the DDF forecast for the fluid height $\zeta(6, 8, t)$. The fluid rest height is $H_0 = 50$m. The time delay parameters here are $D_E = 20, \tau = 17\Delta t$ with $\Delta t = 0.1hr$.



# 5 Addressing Noisy Data

Our twin experiment examples using 'global data' from a shallow water flow have heretofore implicitly assumed we had data with a very high signal to noise ratio. In this section we take these very clean data and add Gaussian noise before constructing a DDF model of the regional dynamics.

We used the 10x10 SWE data and focused on the 3x3 corner region. For each of the 27 time series in this region we added Gaussian noise with zero mean and variance $S\sigma_{\mathbf{O}}^2$. $\sigma_{\mathbf{O}}^2$ is the variance of the signal $\mathbf{O}(\mathbf{R}, t)$ in the sensor region. In this configuration the signal to noise ratio is $S/N = 1/S$ or in dB $S/N = 10 \log_{10}[1/S]$. For small S the data is essentially noise free, as S approaches and exceeds unity, the noise level slowly overcomes the signal.

In Figure (17) we show the y-velocity v(1,1,t), data and DDF Forecast for S = 0.001. In Figure (18) the fluid height $\zeta(1, 1, t)$ for the same noise level is shown as the data and the DDF Forecast.

In Figure (19) we display the the y-velocity $v(1, 1, t))$ data and DDF Forecast when $S = 0.01$, and in Figure (20), the data and DDF Forecast for the fluid height $\zeta(1, 3, t)$ are shown when $S = 0.1$.

As S is increased beyond this, we see significant degradation in the DDF Forecast relative to the data. In Figure (21) S is now unity, and the DDF Forecast, as can be seen has become much less accurate. This noise level corresponds to a signal to noise ratio of 0 dB.

We summarize the robustness to added noise in the data in Figure (22) where the RMS error in the x-velocity is shown for the range of noise level S we considered.



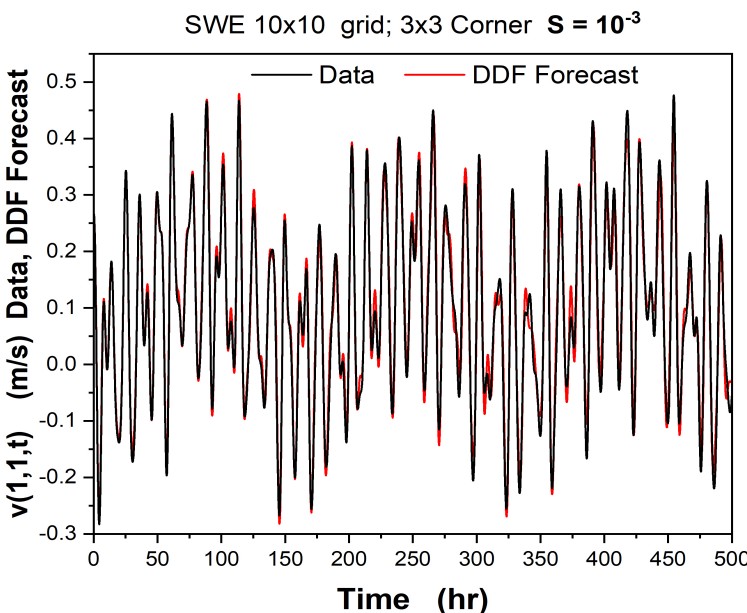

Figure 17: SWE on a $10 \times 10$ grid. The sensor region is comprised of 9 locations, blue dots in Figure (2), in a 3x3 off corner location. We display the y-velocity u(1,1,t) data and DDF forecast. S = 0.001; S/N = 30 dB.

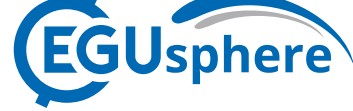

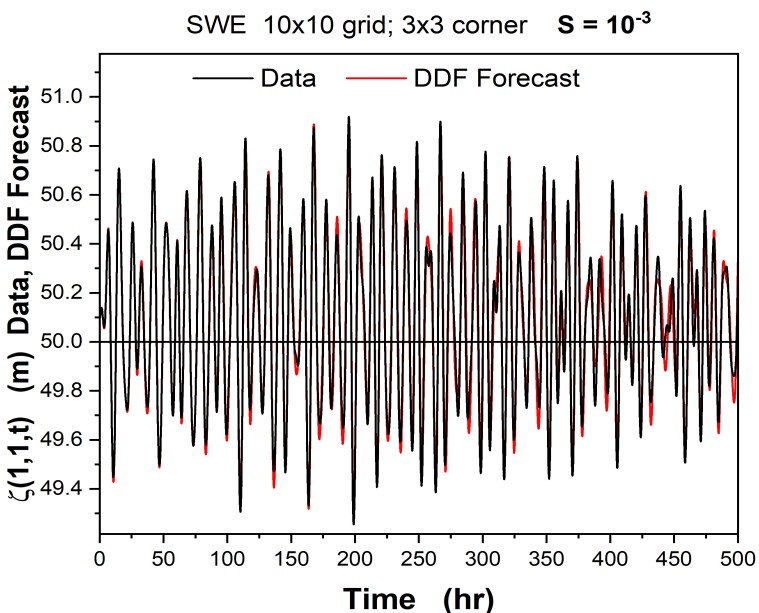

Figure 18: SWE on a $10 \times 10$ grid. The sensor region is comprised of 9 locations, blue dots in Figure (2), in a 3x3 off corner location. We display the fluid height $\zeta(1, 1, t)$ data and DDF forecast. S = 0.001; S/N = 30 dB.



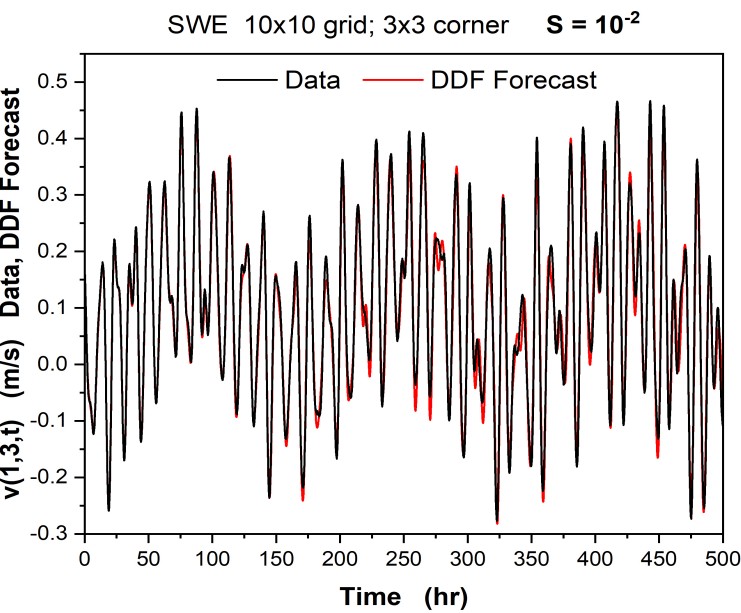

Figure 19: SWE on a $10 \times 10$ grid. The sensor region is comprised of 9 locations, blue dots in Figure (2), in a 3x3 off corner location. We display the y-velocity v(1,3,t) data and DDF forecast. S = 0.01; S/N = 20 dB.





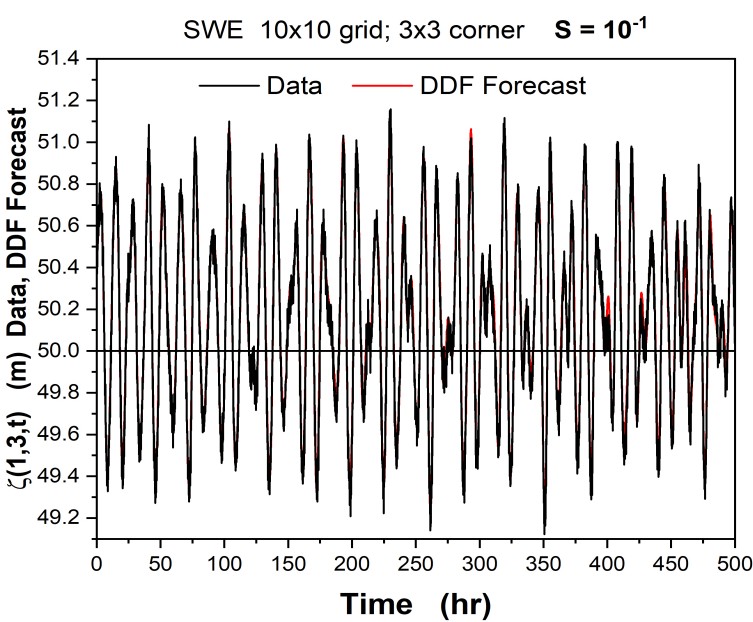

Figure 20: SWE on a $10 \times 10$ grid. The sensor region is comprised of 9 locations, blue dots in Figure (2), in a 3x3 off corner location. We display the xfluid height $\zeta(1,3,t)$ data and DDF forecast. S = 0.1; S/N = 10 dB.





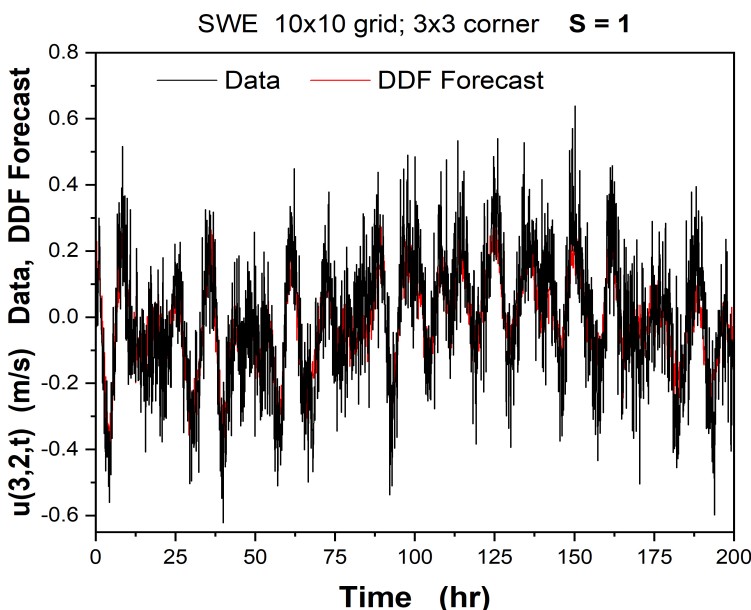

Figure 21: SWE on a $10 \times 10$ grid. The sensor region is comprised of 9 locations, blue dots in Figure (2), in a 3x3 off corner location. We display the x-velocity u(3,2,t) data and DDF forecast. S = 1; S/N = 0 dB. At S = 1 the DDF Forecast has degraded substantially.



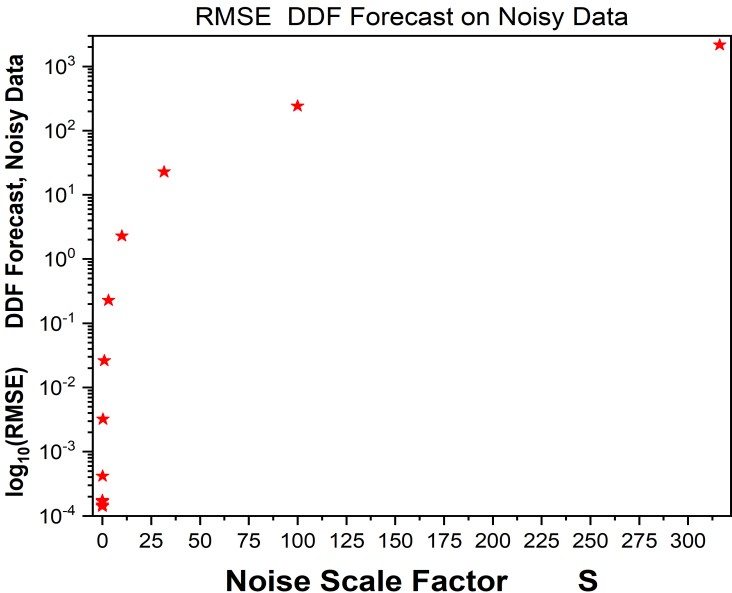

Figure 22: SWE on a $10 \times 10$ grid. The sensor region is comprised of 9 locations, blue dots in Figure (2), in a 3x3 off corner location. We display the RMS error in the x-velocity as a function of S over the range we considered.





# 6 Summary and Discussion

## 6.1 General Remarks

We have introduced a method for building a nonlinear discrete time forecasting system for observed state variables in geophysical dynamical settings where the underlying model of the dynamics is not known. The method relies on observed data to train model parameters in a representation of the unknown dynamical rules. Time series of the data are considered as samples of the distribution in state space visited by the trajectories of the selected physical variables, and the representation of the vector field of the dynamical flow uses well tested interpolating basis functions to give information among the observed samples.

The method for representing the unknown dynamics of the physical flow adopted in our work uses weighted linear combinations of radial basis functions (RBFs) Hardy (1971); Micchelli (1986); Broomhead and Lowe (1988); Schaback (1995); M. D. J. (2002); Buhmann (2009), and the estimation of the weights is a linear algebra problem. This linear algebra operations require a Tikhonov regularization, also known as ridge regression Gruber (1998); Press et al. (2007); van Wieringen (2021). These methods and algorithms implementing them are well tested and widely discussed in the literature.

Selection of observations in a region means that the full state space of the dynamics is not sampled. This entails a projection from the full state space to the subspace of selected observables. Construction of an equivalent state space to the unknown physical state space is, in effect, an **unprojection** from the observed subspace. This is accomplished using time delay embedding widely analyzed in the nonlinear dynamics literature Takens (1981); Eckmann and Ruelle (1985); Sauer et al. (1991); Abarbanel (1996); Kantz and Schreiber (2004).

The Physics of using time delays comes from the fact that the temporal evolution of the observed state variables depends on the full set of (unknown) state variables; see Equation (3) and Equation (4). Over a time delay the full set of state variables is in operation, and the information about the unobserved state variables is conveyed to the observed variables through this feature of the dynamics.

The use of time delay embedding introduces at least two additional quantities to be estimated by data: the value of the time delay $\tau = hT_h$ and the number of required time delays $D_E$. If there are many time scales in the





physical system of interest, many time delays maybe useful along with different numbers of delayed observables $D_E$ Judd and Mees (1995); Hirata et al. (2006).

These considerations led us to formulate the DDF task for forecasting the future of $D_R$ dimensional observed state variables $\mathbf{O}(\mathbf{R}, t)$ located in a spatial region $\mathbf{R}$ as

$$
\begin{aligned}
\mathbf{O}(\mathbf{R}, n+1) = \mathbf{O}(\mathbf{R}, n) + \\
\mathbf{f_R}(\mathbf{TD}(n), \boldsymbol{\chi}) + \frac{h}{2}\left[[\mathcal{F}(\mathbf{R}, n), 0] + [\mathcal{F}(\mathbf{R}, n+1), 0]\right].
\end{aligned} \tag{19}
$$

The $\mathbf{O}(\mathbf{R}, t)$ are $D_R$ dimensional $\mathbf{O}(\mathbf{R}, t) = \{O_\alpha(t)\}$; $\alpha = 1, 2, ..., D_R$. The dynamical map for the regional observables, written in components, is seen to be

$$
\begin{aligned}
O_\alpha(n+1) = O_\alpha(n) + \\
f_\alpha(\mathbf{TD}(n), \boldsymbol{\chi}) + \frac{h}{2}\left[[\mathcal{F}(\mathbf{R}, n), 0] + [\mathcal{F}(\mathbf{R}, n+1), 0]\right]_\alpha.
\end{aligned} \tag{20}
$$

in which $\mathbf{TD}(t_n) = \mathbf{TD}(n)$ are time delay vectors of dimension $D_E D_R$

$$
\mathbf{TD}(t) = [\mathbf{O}(\mathbf{R}, t), \mathbf{O}(\mathbf{R}, t - \tau), \mathbf{O}(\mathbf{R}, t - 2\tau), ..., \mathbf{O}(\mathbf{R}, t - (D_E - 1)\tau)], \tag{21}
$$

and $\boldsymbol{\chi} = \{w_{\alpha q}, c_{\alpha j}, \sigma, B, D_E, T_h\}$; $\tau = h T_h$.

As noted this implies that a representation of the vector field for the flow of each observed quantity is required. We recognize that depending on the number of state variables one chooses to observe and the number of geographical locations at which one observes them, the computational burden required for DDF forecasting may increase substantially. It is unlikely to match the requirements for a full scale regional or global model as presently implemented.

The regional forcing of the system is given as $\mathcal{F}(\mathbf{R}, t_n) = \mathcal{F}(\mathbf{R}, n)$. The external forcing must be given in a DDF formulation just as it is required when we present a detailed physical model of the dynamics Roeckner et al. (2003); Staff (2021).

It is important that the forcing driving the global dynamics enters in an additive fashion in many physical systems, fluid dynamics among them, so the intrinsic properties of the global or regional system is separated from the



forces that drive the system into motion. Once the intrinsic aspects of the dynamics is encoded in the representation of the required flow vector fields, it should respond to any presentation of forcing as does a detailed model of the dynamical system.

## 6.2 The Specific Results in this Paper

In the example of DDF regional forecasting we investigated in this paper we took as our global dynamics the PDEs of shallow water flow realized on a rectangular mid-latitude plane. Clearly this is a useful but major reduction in complexity compared to the collection and set of data from real field observations. We have found it instructive to begin in this manner, and the results illustrate the issues to be encountered in a practical application of the formulation.

In this framework we demonstrated in a number of scenarios that observations of a subset of 'global' shallow water flows can be use to build a discrete time flow $\mathbf{O}(\mathbf{R}, t) \rightarrow \mathbf{O}(\mathbf{R}, t + h)$ allowing for accurate forecasting beyond a temporal domain where data has been previously collected.

We also noted that in the choice of time delay embedding parameters $\{D_E, \tau = T_h h\}$ there is a range of $D_E$ over which excellent forecasting can be achieved.

By adding Gaussian noise to the SWE data, we found a rather robust ability to make accurate DDF Forecasts in a region of the 'global' model until about S = 1, or 0 dB signal to noise.

## 6.3 Looking Forward

In investigating the examples we presented we were required to generate our own data, performing what is often called a *twin experiment* Abarbanel (2022) at a choice of grid points used to approximate solutions to the SWE PDFs. As one proceeds to using our results to guide DDF weather forecasting, at no stage do we approximately solve some physical PDEs on a grid of some spatial resolution, we have no restrictions on where the regional measurements may be performed.

Equally important to note is that the information about the actual dynamics of the overall system of interest are not encoded into physical properties of a model such as effective viscosities at a selected spatial resolution or parametrizations of sub-grid scale dynamics, etc.





The observed information is now located in the estimated parameters $\chi$ of the vector field representations and in the time delay reconstruction of a space equivalent to the full state space of the system. That is, the information in the data remains as it was, but it is redistributed to a different representation of the global dynamics than is encountered in the physical equations of the global problem.

In working with observed data, there are no grid points. These are only introduced to aid in the solution of the PDEs of a physical model. The locations where one sets out sensors are totally up to the users. The discrete time predictive models of observed quantites summarized in Equation (15) are built with out concern for the spatial resolution of grid points in a physical model, without specifying the dynamics of subgrid scale quantities, such as cloud dynamics, and setting aside similar concerns in the formulation of detailed models of the atmosphere and ocean.

As we pointed out earlier, the gains in using DDF models, must be recognized as having been achieved by relinquishing many of the details about earth systems processes that are captured by detailed physically based PDEs for earth systems processes. In DDF only the observed state variables can be forecast. When that is what actually wishes to know, it provides another way to sample data on observed quantities and forecast them.

# Acknowledgements

We acknowledge support the Office of Naval Research, Grants N00014-20-1-2580 and N00014-19-1-2522. Discussions with Erik Bollt, Daniel Gauthier, and Steve Penny were central to pursuing this work.



# 7  Appendix

## 7.1  Performing DDF on Observed Data

The purpose of this section is to provide the reader with more detail on how the implementation of DDF works in practice and to identify what steps would need to be taken by the reader in order to implement DDF to fit their own data, observed or numerically generated for a twin experiment. Clearly the first step is to collect the data, and we assume that is where we start.

In order to implement DDF some choices must be made, the first choice will be in selecting a function representation of the flow vector field. We have tested Taylor Series Representations, RBF's, and RBF's plus polynomial. While the best choice will be dependent upon the system being studied, we've found general success with various representations.

In the coming sections we'll discuss different options and offer some guidance on choosing a representation for a data set. After choosing a function representation, we must choose a way to select centers for the RBF's, choose an appropriate time delay embedding dimension and time delay, and finally perform a hyper parameter search.

## 7.2  Designing a Representation of the Discrete Time Flow Vector Field

To start every problem in DDF we must first choose a function representation, $\mathbf{f}(\mathbf{S}(\mathbf{r}, t_n), \boldsymbol{\chi})$, for the data set we are studying. The way we choose to represent vector fields for a discrete time map will have the largest impact on the predictive power of DDF, we've found that this choice carries far more weight than a good choice of hyper-parameters or an apt selection of centers.

This paper has only focused on the use of RBF's, specifically the Gaussian RBF plus the first order polynomial of each observed dimension. For the shallow water flow data we found that using using polynomial terms in our $\mathbf{f}(\mathbf{S}(\mathbf{r}, t_n), \boldsymbol{\chi})$ representation to match the polynomial terms in the actual SWEs and letting the Gaussian RBF's capture the rest of the behavior of the SWEs works quite well.

We will go on to discuss more details of RBF's, we want to note that other choices exist; these choices include but are not limited to Multilayer Perceptrons, Hermite Interpolation, or any other numerical tool for interpolation. One item to keep in mind is that the RBF expansion is linear in the





RBF weights, allowing us to use linear algebra to estimate them.

### 7.2.1   Choosing a Radial Basis Function

The Radial Basis Function (RBF) was invented by Hardy Hardy (1971) for
interpolation among observed samples of a function of multivariate variables.
There is now an extensive literature on RBF's and their effectiveness as
interpolation functions. In this literature, many different choices for the
form of the RBF have been found to work Hardy (1971); Micchelli (1986);
Broomhead and Lowe (1988); Schaback (1995); M. D. J. (2002); Buhmann
(2009).

The parameters in our choice of RBF $\mathbf{f}(\mathbf{TD}(r,n),\boldsymbol{\chi})$, appearing in Equa-
tion (14), are estimated as follows

The $\mathbf{O}(\mathbf{R},t)$ are $D_R$ dimensional $\mathbf{O}(\mathbf{R},t) = \{O_\alpha(t)\};\ \alpha = 1,2,...,D_R$.
The dynamical map for the regional observables is

$$O_\alpha(n+1) = O_\alpha(n) +$$
$$f_\alpha(\mathbf{TD}(n),\boldsymbol{\chi}) + \frac{h}{2}\bigg[[\mathcal{F}(\mathbf{R},n),0] + [\mathcal{F}(\mathbf{R},n+1),0]\bigg]_\alpha$$
$$\mathbf{TD}(t) = [\mathbf{O}(\mathbf{R},t),\mathbf{O}(\mathbf{R},t-\tau),\mathbf{O}(\mathbf{R},t-2\tau),...,\mathbf{O}(\mathbf{R},t-(D_E-1)\tau].(22)$$

To estimate the parameters $\boldsymbol{\chi}$ we minimize the objective function with
respect to $\boldsymbol{\chi}$.

$$C(\boldsymbol{\chi}) = \sum_{N_c+1}^{N_T}\bigg\{[\mathbf{O}(n+1) - \mathbf{O}(n) - \mathbf{f}(\mathbf{TD}(n),\boldsymbol{\chi})$$
$$-\frac{h}{2}\bigg[[\mathcal{F}(\mathbf{R},n),0] + [\mathcal{F}(\mathbf{R},n+1),0]\bigg]\bigg\}^2. \tag{23}$$

In Equation (23)

$$f_\alpha(\mathbf{TD}(n),\boldsymbol{\chi}) = \sum_{q=1}^{N_c} w_{\alpha q}\psi((\mathbf{TD}(n) - \mathbf{TD}^c(q))^2,\sigma) + \sum_{j=1}^{D_R} c_{\alpha j}\mathbf{O}_j(n) \tag{24}$$

Even though we can choose any RBF for $\boldsymbol{\psi}$, the $C(\boldsymbol{\chi})$ is always linear in
the weights $w_{\alpha q}, c_{\alpha j}$, enabling the use of the linear algebra of Ridge Regression
or Tikhonov regularization in estimating them.





In working with our example of the SWE, we found that the Gaussian
RBF $\boldsymbol{\psi}_G((\mathbf{TD}(n) - \mathbf{TD}^c(q))^2$ plus (1st order) polynomials in $\mathbf{TD}(n)$ to be
more effective than $\boldsymbol{\psi}_G((\mathbf{TD}(n) - \mathbf{TD}^c(q))^2$ alone.

To have a simplified notation for solving the regularized linear algebra
problem, let us call

$$\mathbf{Y}(n) = \mathbf{O}(n+1) - \mathbf{O}(n) - \frac{h}{2}\bigg[[\mathcal{F}(\mathbf{R}, n), 0] + [\mathcal{F}(\mathbf{R}, n+1), 0]\bigg], \qquad (25)$$

the vector of RBF coefficients,

$$\mathbf{P} = \{w_{\alpha q}, c_{\alpha j}\}, \qquad (26)$$

and the RBF + Polynomial terms are the matrix $\mathbf{M}$.

Then the regularized cost function is given by Press et al. (2007)

$$C_B(\boldsymbol{\chi}) = \sum_n \bigg[\mathbf{Y}(n) - \mathbf{PM}(n)\bigg]^2 + B\mathbf{P}^T\mathbf{P}. \qquad (27)$$

T indicates the transpose.

The minimization of $C_B(\boldsymbol{\chi})$ with respect to $\mathbf{P}$, gives us the regularized
solution

$$\mathbf{P} = \mathbf{Y} \cdot \mathbf{M}^T \frac{1}{(\mathbf{MM}^T + B\mathbf{I})}. \qquad (28)$$

After we introduce time delay embedding, the DDF parameters are $\boldsymbol{\chi} = \{w_{\alpha q}, c_{\alpha j}, \sigma, B, T_h, D_E\}$.

### 7.2.2 Including Polynomial Terms in Eq. (24)

In this paper we have found the most effective representation for SWE to
be RBF representations with the inclusion of a polynomial term; the poly-
nomial terms are in the $\mathbf{TD}(n)$. We retain only first order terms in this
paper Schaback (1995); M. D. J. (2002).

With a general set of observations where we might have no particular in-
sight into the dynamics, one should add the polynomials present in a general
formulations of the problem Schaback (1995); M. D. J. (2002).





### 7.2.3  How to choose Centers

The choice of centers, $\mathbf{TD}^c(q)$ vector in the RBF, will be dictated by the data points in the training set.

We had initial success taking every $n^{th}$ data point in the training window to be a center, this method is quick, but it may poorly saturate regions of low density and is prone to overlapping centers and to leaving empty space.

We recommend taking the approach of trying to saturate all important areas of space densely with centers to provide DDF with the most accurate update rule in those regions. With this in mind, we have opted to use K-means clustering to choose our centers for us. While there may be more advanced strategies, we have found K-means to work well enough, and we used it in finding all of the results in the present paper Du and Swamy (2006).

Another aspect to consider is the number of centers. The general rule of thumb is that more centers is better, but this comes at the cost of both memory and computational time. For the results in this paper, typically around 1000 centers were used.

One strategy could be to do preliminary testing with fewer centers and increase the number of centers. The fewer centers one can get by with, the better, as they will dramatically improve the computational time using the DDF map over the use of excessive centers.

### 7.2.4  Time Delay Embedding (TDE)

When studying physical systems, it is often very difficult or completely impossible to measure all observable quantities of the system. There is information that is lost in these unobserved quantities that must be recaptured through the use of time delay embedding. When studying the subregions of the shallow water flows as global data or some region of the atmosphere/oceans, we take full advantage of TDE in DDF to recapture the dynamical information of the system.

For the purpose of this paper, we took subregions of a grid generated by the SWE, and we required a choice of a time delay and dimension size for TDE.

To find an effective number of time delay dimensions, $D_E$, Taken's Theorem guarantees that the attractor can be reconstructed in as little as $2D_A + 1$ where $D_A$ is the box counting fractal dimension dimension of the original state space. However, by calculating the number of false nearest neigh-



bors Abarbanel (1996) at each $D_E$ value, we can get an estimate of the ideal number of time delay dimensions. As this reconstruction proceeds, we eventually reach a point where the number of false nearest neighbors drops to a negligibly low number. This number of time delay dimensions serves as our estimate for $D_E$.

The time delay $\tau$ is conveniently taken to be a multiple of the observed temporal step h in the data collection. $\tau = T_h h$. Also the embedding dimension $D_E$ for $\mathbf{TD}(t)$ is always an integer. So in minimizing the cost function $C(\boldsymbol{\chi})$ we must search a grid of integers $\{T_h, D_E\}$ along with a search in continuous variables $\{w_{\alpha q}, B, \sigma^2 = \frac{1}{2R}\}$.

We do not need to separately forecast all components of $\mathbf{TD}(t)$. If we forecast only $\mathbf{O}(t)$, the remaining components are forecast as well. We do require the full time delay vectors in the arguments of the RBF.

### 7.2.5   Finding the hyper-parameters R and B in $\chi$

When performing DDF we are faced with the choice of picking a good value for hyper-parameters that affect our training and forecasting. For the case of the Gaussian RBF we have to pick an R value for the coefficient in the exponent of the RBF and a B value for the regularization term in Ridge Regression. To find a good value of B and R, it is typically easiest to perform the brute force method of grid searching across orders of magnitude of R and B; once a good result is found, further sweeps can be conducted with greater precision if greater accuracy is desired. Alternatively, one method the reader could use would be to use a genetic algorithm to find a precise result after performing a wide grid sweep (this is discussed in the Future Improvements to DDF section under Differential Evolution).

It is the experience of the authors that if a large grid sweep across many orders of magnitude of R and B fail to provide any good results that the next best course of action would be to take another look at the function representation and try something new. We typically start with a basic order of magnitude sweep for both B and R starting both values at 1e-8 and going up to 1e+1 (the best value for B and R are very data set dependent, so initial large sweeps are always necessary). If the function representation is not working well, or is a bad fit for the data, then our experience has taught us that it is a fruitless endeavour to keep grinding away at testing more and more hyper-parameter values.





## 7.3 Programming DDF

We did all of our testing and programming in Python. Here we will offer
a few, hopefully, helpful suggestions for the reader in performing their own
DDF calculations as well as provide sample code.

### 7.3.1 PreBuilt DDF Code in Python

Here is a link to the code we used (written and tested in Python) that
have been published in GitHub. It includes the python scripts and Jupyter
Notebooks used to get the results in this paper, it also includes a few examples
as well as some information as to how we solve the SWE to generate the data
we use Clark

### 7.3.2 Memory Management

An important note to consider is the size of the matrices involved in training,
for they can grow to the size of gigabytes. For example, the largest matrix
involved in training is the $\mathbf{M}$ matrix consisting of all values of the RBF's at
all times, an N by $N_c$, can have a length of up to 25,000 data points with as
many as 5,000 centers. Typically we use float64 for all our values resulting
this matrix using $25,000 * 5,000 * 8/1e9 = 1$ Gigabyte, then we'll need another
gigabyte for its transpose. This could be a limiting factor if one is running
multiple tests in parallel on a CPU or on a cluster with limited storage.

### 7.3.3 Parallelizing DDF

This section serves less as a guide and more as a suggestion and an obvious
disclaimer to take advantage of the ability to run DDF code in parallel. The
grid searching method discussed in the "Finding Hyper Parameters" section
of the appendix lends itself very readily to parallel programming. Even the
Differential Evolution method that is discussed in the Future Improvements
to DDF section can have it's trials ran in parallel. Note that a single trial of
DDF doesn't have much potential for parallel operations as the training is
just matrix multiplication and the forecasting is a step by step process that
relies of the result of the previous step's calculation.



## 7.4 Future Improvements to DDF

DDF isn't perfect and there is still much left to be explored with this method. There are still many RBF's that we never tested that could potentially work better than the Gaussian we've had success with. There are also smarter ways of choosing hyper-parameters other than brute force grid searching, such as using genetic algorithms like Differential Evolution.

### 7.4.1 Centers and RBF's

We took for granted that the obvious choices for DDF worked out as well as they did, choosing Gaussian RBF's and K-means clustering to put together our DDF experiments were ideas commonly used in the RBF literature Wu et al. (2012); however, they are far from the only good ideas there and others could possibly work better.

The centers could also be chosen by emphasizing places where the first or second derivative are greatest; this idea could work well because it could saturate regions where not many data points exist. Sánchez A (1995)

### 7.4.2 Differential Evolution

Differential Evolution is a genetic algorithm that could be implemented into DDF to improve our ability to create DDF models of observed systems by more precisely picking out sets of hyper parameters. It works by initiating a parent set of hyper-parameters from a user defined uniform distribution. From there, new "children" sets of hyper parameters are made from combining the parents in a algorithmic way and comparing the new forecast that is made to the old one. A user defined cost function is used to compare the the children to their parents, as generations go by, parents will gradually be replaced by children until all the parents converge on a minimum in the cost function. If the user is able to define a clever cost function, they could get a result that hope to surpass those from simple grid searching. The full method is described in  Storn and Price (1997).



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
