# Peer review of "Data Driven Regional Weather Forecasting"

_EGUsphere, 2022_

## Referee Comment (RC1)

**Peer-Review of**
**"Data Driven Regional Weather Forecasting"**

November 30, 2022

**1 Summary of the Content**

The objective of this paper is to provide a data-driven forecast method to replace the conventional model forecast. The authors use the time-delayed observations to reduce the uncertainty of data-driven forecast, and then set up a parameterization for the forecast model. The parametrization framework consists of a polynomial term and a analog-type of method. Once the parameterization is set, the parameters are estimated using standard methods. The authors tested this method using a shallow water equation on a 10×10 grid. The authors considered several different observed regions. The overall numerical performance is quite good.

**2 Overall Feedback**

The general idea is easy to understand and the numerical results look good and clear. However, certain aspects still need to be clarified. I recommend major revision for this manuscript.

**3 Comments**

**1) Lines 580-584, 622-628**

In lines 580-584, the authors claim that the polynomial term is used to represent the polynomial terms in the original dynamics. SWE has nonlinear terms, so the polynomials should have degree larger than 1. However, in lines 622-628, the authors seem suggest that the choice of linear polynomial is due to their insight about the dynamics of SWE. This contradicts to their reasoning in lines 580-584. Please explicitly explain why degree 1 polynomials are good choices for SWE, and what is your insight into the dynamics of SWE.

**2) I recommend the authors provide their realization of their algorithm with the paper. This would make introspection/adaption/expansion of their results by the community much easier and faster.**

**3) Please add some figures to visualize your choice of "centers", i.e. the data at which of the time steps are used in the RBFs?**

**4) The scientific story of this manuscript needs to be justified with more reasoning and tests.**

I have another interpretation of the methodology presented in this manuscript. The use of "centers" in this method essentially makes use of the ergodic nature of the SWE with forcing. The "centers" are sample points on the attractor of the system. The Gaussian RBFs serve as a weight functionals and the coefficients $w_{\alpha q}$ are approximately the true dynamics corresponding to the state $\mathbf{TD}^c(q)$. In this case, the polynomial term could serve as a residual term. Can the authors quantify the contributions from the polynomial term and the RBF term, and discuss about the results?

**5) Throughout the manuscript the authors use $\mathcal{R}$ to denote the set of real numbers. Conventionally, people use $\mathbb{R}$ instead of $\mathcal{R}$.**

**6) Some of the notation/index in the equations are confusing.**

For instance, In Eq.(11), you probably need a subscript $a$ in $\mathbf{S}(n)$ and $\mathbf{S}(n+1)$. At the end of Eq.(17), what does $\mathbf{O}_j(n)$ mean?

---

## Referee Comment (RC2)

**Peer-Review of**
**"Data Driven Regional Weather Forecasting"**

December 2, 2022

**1 Comments**

**1) About the role of time-delayed embedding**

Let me elaborate more on what I meant. Taken's embedding theorem says that, under certain conditions, there exists an integer $D_E$ so that $\mathbf{TD}(n)$ **DETERMINES** some $\mathbf{TD}(n+1)$. There exists a smallest such integer, which we denote by $D_E^{\max}$. Then for $D_E < D_E^{\max}$, there might be several different values of $\mathbf{TD}(n+1)$ that follows the same $\mathbf{TD}(n)$. In this case, the forecast system has uncertainty. And intuitively the more lags you use, the less uncertainty you would have. This is what I mean by "the authors use time-delayed observations to reduce the uncertainty of the forecast system". For instance, in Lorenz 63 system, suppose you observe the time series $x(t)$ from the 3-dimensioanly state vector time series $(x(t), y(t), z(y))$. For $D_E = 2$, numerically you still can construct a forecast system by parameterization and whatever method you use to estimate the parameters. But in fact there does not exist a deterministic function $f$, so that $f(\mathbf{TD}(n)) = \mathbf{TD}(n+1)$. Therefore no matter how you parameterize your forecast system, you can only get a "mean value" instead of the exact value of the future.

Theoretically, Taken's embedding theorem requires the dynamical system to be finite dimensional. This means that it must be a finite dimensioanl ODE or a PDE that has a finite dimensional inertial manifold. It is likely that the SWE with dissipation is the second case. But there is no demonstration about this point in this manuscript (and similarly in many others). Without a rigorous demonstration on whether the dynamical system is essentially finite dimensional, "Taken's embedding theorem" is merely a "faith" but might not be the fact. In this case, the time-delayed embedding technique is expected to reduce the uncertainty of the forecast system, instead of providing a deterministic and completely accurate forecast system.

It is true that your observations are taken from the 10×10 grid simulation of SWE, which in fact is a finite dimensional ODE. But this also implies that your obs are not taken from the true SWE dynamical system. For your interest, you can try increasing the resolution of your grid and do the same experiment for the same observed locations (the $3 \times 3$ sub-grid in the $10 \times 10$ grid). Maybe you will find that larger $D_E$ is needed in order to provide an accurate forecast. If this is not obvious in your experiment, you can try using a smaller dissipation rate ($A$ and $\epsilon$) and a higher resolution.

As written in the introduction of this manuscript, the core idea of this manuscript is to provide a data-driven method to forecast the future of the observed time-series from the true nature (i.e. the infinite dimensional SWE). Therefore I think it is more reasonable to suggest in this manuscript that the time-delayed embedding is used to reduce the uncertainty of the forecast model, instead of providing a diffeomorphism between $S$ and $\mathbf{TD}(n)$.

**2) About the role of RBFs**

Thanks the author for pointing out that the paragraph after Eq.(7) is equivalent to the interpretation of the reviewer, which I indeed did not notice. What I am curious about is that which term, the RBFs or the polynomial term, contributes more in the forecast model. It would be more clear if the authors can provide some numerical results on this. And, again, what is the rationale behind the choice of degree 1 polynomial? Is it due to some insight of the dynamics of the SWE or merely result-driven?

---

## Referee Comment (RC3)

**REVIEW DATA DRIVEN REGIONAL WEATHER FORECASTING**

This paper uses data driven techniques to forecast partially observed systems. The proposed framework relies on phase space reconstruction and defines an approximate discrete vector field on the reconstructed phase space to forecast the regional observations. The model was tested on a Shallow Water Equation setting with various configurations of the observations. Overall, the paper is well written and suitable for the audience of NPG. However, several aspects need some investigation to further strengthen the paper.

Major comments :

1. Positioning of the paper and bibliography: I liked your text in the 1.1 section, especially the shift in the machine learning community to account for physical constraints in the models. However, your state-of-the-art section is missing two crucial aspects. You did not discuss the related works on machine learning for regional forecasting systems (for example: [1, 2, 3, 4]). These references do not have explicit titles like yours, but are very similar in spirit. Furthermore, you also need to discuss the state of the art in data driven modeling for partially observed systems (for example: old traditional methods [5], new ones [6, 7, 8, 9, 10]). These works do use phase space reconstruction techniques, as you do for predicting partially observed systems.

2. Notation: overall I found the paper very smooth and pleasant to read, except when you introduced the models equations. I think there is a way to simplify the notations to make the paper even smoother. For example, you use for the "plane coordinates" the variable $\mathbf{r} = \{x, y\}$ which is equal to $\{r_1, r_2\}$ which is equal to $\{r_{10} + i\Delta x, r_{20} + j\Delta x\}$ these are lots of symbols to keep track of. I think you can only keep $\mathbf{r} = \{r_{10} + i\Delta x, r_{20} + j\Delta x\}$ and explain what is $\Delta x, \Delta x$ and $i, j$. Also, you use both $\Delta t$ and $h$ for the time step. You can also explain that you use capital letters for the observations and regular letters for the states. In equations (3), (4) ...etc. when you write that

$$\frac{d\mathbf{S}(i,j,t)}{dt} = \mathbf{F}_{i,j}(\mathbf{S}(i,j,t),\theta) + [\mathcal{F}(i,j,t),0]$$

I understand that the state $\mathbf{S}(i,j,t)$ at the grid point $i,j$ depends only on $\mathbf{S}(i,j,t)$ and the forcing. In order to avoid confusion, make the above equation depend on a defined global state, for example $\mathbf{S}(t) = [\mathbf{S}(0,0,t), \ldots, \mathbf{S}(n_x, n_y, t)]^T \in \mathbb{R}^{3(n_x \times n_y)}$. Also, make sure the forcing notation is consistent with the definition of the global state vector $\mathbf{S}(t)$.

3. Regarding the optimization of the embedding parameters: after reading lines 347 to 352, I don't really understand how you parameterized your embedding parameters. In the appendix, it says that you use FNN to get the dimension, but it's not clear how you calibrate the delay. Please further explain your methodology here.

4. On the delay embedding parameterization: You state in lines 379 to 382 that in the Takens delay embedding theorem, there is no underlying assumption on the time delay. I have an example that contradicts this. Imagine that your limit-set is periodic with period T and that you use a time delay of T. In this situation, your reconstruction will not be diffeomorphic to the underlying limit set. Actually, this is written the other way around in the Fractal Delay Embedding Prevalence Theorem (Theorem 2.5 in [11]).

5. Benchmark: This is extremely important. You should provide a benchmark of your results to help readers better understand the performance of your work. At least compare to persistence, but I would recommend testing against other state-of-the-art machine learning models.

6. Presentation of the experiments: For the experimental section, I recommend including only the sensors configuration figure and a single forecasting figure per experiment. The remaining figures should be placed in the appendix.

7. On the dimension of the embedding: You state in lines 384 to 388 that the result of Sauer et al. indicates that if $D_E > 2D_A + 1$ the unprojection would work. I think that this result is from Whitney Embedding Prevalence Theorem (theorem 2.2 in [11]). Also, when you assume that $D_A$ is near 300 (lets say $D_A = 300$) then $D_E \approx 10, 20$ is very far from satisfying the condition $D_E > 2D_A + 1$. Please rectify this sentence.

8. On the scalability of the model: In your experiments, your initial state is of dimension 300. However, the embedding you used is of dimension $27 * 10$ to $27 * 20$. On higher dimensional systems as you mensioned, the dimension is much larger that 300 ($\mathbf{O}(10^6)$ for example). Do you expect your model to be in even larger dimensions than $\mathbf{O}(10^6)$. If it is the case, what is the cost of the model ? If not, why would you expect to get a smaller dimension.

Minor comments,

1. Do you mean embedding when you write unprojection ? If so, use embedding.

2. in lines 100 to 101, It's good to explain what one loses when bypassing equations.

3. in lines 96 to 97, I don't understand why you avoid uncertainties in the data. If your model is optimized from data and there is some uncertainty in the data, you would have uncertainty in the models.

4. I found the sentence from line 253 to 257 difficult to read.

5. Equation 6 has two commas.

I hope that the authors will understand my comments in a constructive way, and that I value their work and the time they invested in the preparation of the manuscript. It might be that I have misunderstood something, in this case, if something wasn't clear for me as a reviewer, it is possible that it wouldn't be clear also for the readers.

**References**

[1] James Knighton, Geoff Pleiss, Elizabeth Carter, Steven Lyon, M Todd Walter, and Scott Steinschneider. Potential predictability of regional precipitation and discharge extremes using synoptic-scale climate information via machine learning: An evaluation for the eastern continental united states. *Journal of Hydrometeorology*, 20(5):883–900, 2019.

[2] Veronica Nieves, Cristina Radin, and Gustau Camps-Valls. Predicting regional coastal sea level changes with machine learning. *Scientific Reports*, 11(1):1–6, 2021.

[3] Changjiang Xiao, Nengcheng Chen, Chuli Hu, Ke Wang, Zewei Xu, Yaping Cai, Lei Xu, Zeqiang Chen, and Jianya Gong. A spatiotemporal deep learning model for sea surface temperature field prediction using time-series satellite data. *Environmental Modelling & Software*, 120:104502, 2019.

[4] Serkan Kartal. Assessment of the spatiotemporal prediction capabilities of machine learning algorithms on sea surface temperature data: A comprehensive study. *Engineering Applications of Artificial Intelligence*, 118:105675, 2023.

[5] Henry D. I. Abarbanel and Upmanu Lall. Nonlinear dynamics of the great salt lake: system identification and prediction. *Climate Dynamics*, 12(4):287–297, Mar 1996.

[6] Ahmad Kazem, Ebrahim Sharifi, Farookh Khadeer Hussain, Morteza Saberi, and Omar Khadeer Hussain. Support vector regression with chaos-based firefly algorithm for stock market price forecasting. *Applied Soft Computing*, 13(2):947 – 958, 2013.

[7] Jordan Frank, Shie Mannor, and Doina Precup. Activity and gait recognition with time-delay embeddings. In *Proceedings of the Twenty-Fourth AAAI Conference on Artificial Intelligence*, AAAI'10, pages 1581–1586. AAAI Press, 2010.

[8] S Ouala, D Nguyen, L Drumetz, B Chapron, A Pascual, F Collard, L Gaultier, and R Fablet. Learning latent dynamics for partially observed chaotic systems. *Chaos: An Interdisciplinary Journal of Nonlinear Science*, 30(10):103121, 2020.

[9] Georg A Gottwald and Sebastian Reich. Combining machine learning and data assimilation to forecast dynamical systems from noisy partial observations¡? a3b2 show [editpick]?¿. *Chaos: An Interdisciplinary Journal of Nonlinear Science*, 31(10):101103, 2021.

[10] Charles D Young and Michael D Graham. Deep learning delay coordinate dynamics for chaotic attractors from partial observable data. *arXiv preprint arXiv:2211.11061*, 2022.

[11] Tim Sauer, James A. Yorke, and Martin Casdagli. Embedology. *Journal of Statistical Physics*, 65(3):579–616, Nov 1991.

---

## Community Comment (CC1)

Comments on Review by Referee Number 1

We are most appreciative of the detailed and prompt review of our paper "Data Driven Weather Forecasting." We note here some comments related to the review.

We understand that the review period is not complete, and we will address all posts related to our paper and provide a revised manuscript at that time. There are some items which we wish to address at this time:

The reviewer wrote:

1. "The authors use the time-delayed observations to reduce the uncertainty of data-driven forecast."

   This is not the role of time delay embedding.

   As discussed in detail in Section 2.2, see Equations (4), (5) and (6), and the text around them, the equations (4) for the observed subset of the state variables $S(r,t)$, which we call $O(R,t)$, depends on $S(r,t)$, and the core of Takens' theorem is the the time delay space with state vectors TD, in Equation (12) is an equivalent space, formally the spaces with vectors S and TD are connected by a diffeomorphism. We do not know, from observations alone the vectors S, but we do know the vectors TD. The use of time delay vectors in Equation (13) and elsewhere is dictated by the structure of the differential equations generating the data, even though we do not know the vector fields in Equation (4).

   Please note that the TD vectors have components which are the observed $O(R,t)$ and its time delays.

2. "I have another interpretation of the methodology presented in the manuscript. The use …"

   Of course, we agree with this paragraph, and it is clearly stated in Section 2.3.1 in the full paragraph after Equation (7) about the use of RBF's or any other representation of the flow vector field: "We can think of the observed samples as points of information about distribution $f(S,\chi)$ and ask that the representation give us an interpolating function among the observed point locations $S(r,t_n) = S(r,n)$."

   This is totally equivalent to the "another interpretation" of the reviewer. We will clarify our language in the revised manuscript.

Our thanks again for the review and the suggestions on clarifying the language we have used.

---

## Author Comment (AC1)

Referee Number 1:

**1 Summary of the Content**

The objective of this paper is to provide a data-driven forecast method to replace the conventional model forecast. The authors use the time-delayed observations to reduce the uncertainty of data-driven forecast, and then set up a parameterization for the forecast model. The parametrization framework consists of a polynomial term and a analog-type of method. Once the parameterization is set, the parameters are estimated using standard methods. The authors tested this method using a shallow water equation on a 10×10 grid. The authors considered several different observed regions. The overall numerical performance is quite good.

**2 Overall Feedback**

The general idea is easy to understand and the numerical results look good and clear. However, certain aspects still need to be clarified. I recommend major revision for this manuscript.

**3 Comments**

**1) Lines 580-584, 622-628**

In lines 580-584, the authors claim that the polynomial term is used to represent the polynomial terms in the original dynamics. SWE has nonlinear terms, so the polynomials should have degree larger than 1. However, in lines 622-628, the authors seem suggest that the choice of linear polynomial is due to their insight about the dynamics of SWE. This contradicts to their reasoning in lines 580-584. Please explicitly explain why degree 1 polynomials are good choices for SWE, and what is your insight into the dynamics of SWE.

**2) I recommend the authors provide their realization of their algorithm with the paper. This would make introspection/adaption/expansion of their results by the community much easier and faster.**

**3) Please add some figures to visualize your choice of "centers", i.e. the data at which of the time steps are used in the RBFs?**

**4) The scientific story of this manuscript needs to be justified with more reasoning and tests.**

I have another interpretation of the methodology presented in this manuscript. The use of "centers" in this method essentially makes use of the ergodic nature of the SWE with forcing. The "centers" are sample points on the attractor of the system. The Gaussian RBFs serve as a weight functionals and the coefficients $w_{\alpha q}$ are approximately the true dynamics corresponding to the state $\mathbf{TD}^c(q)$. In this case, the polynomial term could serve as a residual term. Can the authors quantify the contributions from the polynomial term and the RBF term, and discuss about the results?

**5) Throughout the manuscript the authors use $\mathcal{R}$ to denote the set of real numbers. Conventionally, people use $\mathbb{R}$ instead of $\mathcal{R}$.**

**6) Some of the notation/index in the equations are confusing.**

For instance, In Eq.(11), you probably need a subscript $a$ in $\mathbf{S}(n)$ and $\mathbf{S}(n+1)$. At the end of Eq.(17), what does $\mathbf{O}_j(n)$ mean?

Our Response:

Comments on Review by Referee Number 1

We are most appreciative of the detailed and prompt review of our paper "Data Driven Weather Forecasting." We note here some comments related to the review.

We understand that the review period is not complete, and we will address all posts related to our paper and provide a revised manuscript at that time. There are some items which we wish to address at this time:

The reviewer wrote:

1. "The authors use the time-delayed observations to reduce the uncertainty of data-driven forecast."

   This is not the role of time delay embedding.

   As discussed in detail in Section 2.2, see Equations (4), (5) and (6), and the text around them, the equations (4) for the observed subset of the state variables $S(r,t)$, which we call $O(R,t)$, depends on $S(r,t)$, and the core of Takens' theorem is the the time delay space with state vectors TD, in Equation (12) is an equivalent space, formally the spaces with vectors S and TD are connected by a diffeomorphism. We do not know, from observations alone the vectors S, but we do know the vectors TD. The use of time delay vectors in Equation (13) and elsewhere is dictated by the structure of the differential equations generating the data, even though we do not know the vector fields in Equation (4).

   Please note that the TD vectors have components which are the observed $O(R,t)$ and its time delays.

2. "I have another interpretation of the methodology presented in the manuscript. The use …"

   Of course, we agree with this paragraph, and it is clearly stated in Section 2.3.1 in the full paragraph after Equation (7) about the use of RBF's or any other representation of the flow vector field: "We can think of the observed samples as points of information about distribution $f(S,\chi)$ and ask that the representation give us an interpolating function among the observed point locations $S(r,t_n) = S(r,n)$."

   This is totally equivalent to the "another interpretation" of the reviewer. We will clarify our language in the revised manuscript.

Our thanks again for the review and the suggestions on clarifying the language we have used.

Referee Number 1 Had this to say in Response:

**1  Comments**

**1) About the role of time-delayed embedding**

Let me elaborate more on what I meant. Taken's embedding theorem says that, under certain conditions, there exists an integer $D_E$ so that $\mathbf{TD}(n)$ **DETERMINES** some $\mathbf{TD}(n+1)$. There exists a smallest such integer, which we denote by $D_E^{\max}$. Then for $D_E < D_E^{\max}$, there might be several different values of $\mathbf{TD}(n+1)$ that follows the same $\mathbf{TD}(n)$. In this case, the forecast system has uncertainty. And intuitively the more lags you use, the less uncertainty you would have. This is what I mean by "the authors use time-delayed observations to reduce the uncertainty of the forecast system". For instance, in Lorenz 63 system, suppose you observe the time series $x(t)$ from the 3-dimensioanly state vector time series $(x(t), y(t), z(y))$. For $D_E = 2$, numerically you still can construct a forecast system by parameterization and whatever method you use to estimate the parameters. But in fact there does not exist a deterministic function $f$, so that $f(\mathbf{TD}(n)) = \mathbf{TD}(n+1)$. Therefore no matter how you parameterize your forecast system, you can only get a "mean value" instead of the exact value of the future.

Theoretically, Taken's embedding theorem requires the dynamical system to be finite dimensional. This means that it must be a finite dimensioanl ODE or a PDE that has a finite dimensional inertial manifold. It is likely that the SWE with dissipation is the second case. But there is no demonstration about this point in this manuscript (and similarly in many others). Without a rigorous demonstration on whether the dynamical system is essentially finite dimensional, "Taken's embedding theorem" is merely a "faith" but might not be the fact. In this case, the time-delayed embedding technique is expected to reduce the uncertainty of the forecast system, instead of providing a deterministic and completely accurate forecast system.

It is true that your observations are taken from the 10×10 grid simulation of SWE, which in fact is a finite dimensional ODE. But this also implies that your obs are not taken from the true SWE dynamical system. For your interest, you can try increasing the resolution of your grid and do the same experiment for the same observed locations (the $3 \times 3$ sub-grid in the $10 \times 10$ grid). Maybe you will find that larger $D_E$ is needed in order to provide an accurate forecast. If this is not obvious in your experiment, you can try using a smaller dissipation rate ($A$ and $\epsilon$) and a higher resolution.

As written in the introduction of this manuscript, the core idea of this manuscript is to provide a data-driven method to forecast the future of the observed time-series from the true nature (i.e. the infinite dimensional SWE). Therefore I think it is more reasonable to suggest in this manuscript that the time-delayed embedding is used to reduce the uncertainty of the forecast model, instead of providing a diffeomorphism between $S$ and $\mathbf{TD}(n)$.

**2) About the role of RBFs**

Thanks the author for pointing out that the paragraph after Eq.(7) is equivalent to the interpretation of the reviewer, which I indeed did not notice. What I am curious about is that which term, the RBFs or the polynomial term, contributes more in the forecast model. It would be more clear if the authors can provide some numerical results on this. And, again, what is the rationale behind the choice of degree 1 polynomial? Is it due to some insight of the dynamics of the SWE or merely result-driven?

Current Response:

1) **The Time delay theorem tells one how to construct an equivalent space for the dynamics of the source of the observed signal(s). If one does not work in the equivalent space, then either the space is too small and there are unacceptable false nearest neighbors or, if the space is too large, one is wasting time and emphasizing noise in the "extra" dimensions.**

2) **The general construction of Powell, Shaback and others includes polynomial contributions as well as RBF contributions. We have found that the nonlinear terms in the underlying dynamics are well represented by the RBFs and using polynomial terms often yields better forecasting ability. The choice for using first order polynomials comes from the theme in DDF of matching the terms in the update rule to the dynamical equations for improved forecasting power; in this case we use the first order polynomials to capture the first order polynomial behavior and use the RBFs to capture the remaining behavior of the SWE dynamics. Testing has confirmed for us that the inclusion of the first order polynomials does increase predictive power and reduce errors (our intuition for this is that with their inclusion there is less "information" or dynamics for the RBFs to have to capture enabling them to perform better on the remaining dynamics).**

**REVIEW DATA DRIVEN REGIONAL WEATHER FORECASTING**

This paper uses data driven techniques to forecast partially observed systems. The proposed framework relies on phase space reconstruction and defines an approximate discrete vector field on the reconstructed phase space to forecast the regional observations. The model was tested on a Shallow Water Equation setting with various configurations of the observations. Overall, the paper is well written and suitable for the audience of NPG. However, several aspects need some investigation to further strengthen the paper.

Major comments :

1. Positioning of the paper and bibliography: I liked your text in the 1.1 section, especially the shift in the machine learning community to account for physical constraints in the models. However, your state-of-the-art section is missing two crucial aspects. You did not discuss the related works on machine learning for regional forecasting systems (for example: [1, 2, 3, 4]). These references do not have explicit titles like yours, but are very similar in spirit. Furthermore, you also need to discuss the state of the art in data driven modeling for partially observed systems (for example: old traditional methods [5], new ones [6, 7, 8, 9, 10]). These works do use phase space reconstruction techniques, as you do for predicting partially observed systems.

2. Notation: overall I found the paper very smooth and pleasant to read, except when you introduced the models equations. I think there is a way to simplify the notations to make the paper even smoother. For example, you use for the "plane coordinates" the variable $\mathbf{r} = \{x, y\}$ which is equal to $\{r_1, r_2\}$ which is equal to $\{r_{10} + i\Delta x, r_{20} + j\Delta x\}$ these are lots of symbols to keep track of. I think you can only keep $\mathbf{r} = \{r_{10} + i\Delta x, r_{20} + j\Delta x\}$ and explain what is $\Delta x, \Delta x$ and $i, j$. Also, you use both $\Delta t$ and $h$ for the time step. You can also explain that you use capital letters for the observations and regular letters for the states. In equations (3), (4) ...etc. when you write that

$$\frac{d\mathbf{S}(i,j,t)}{dt} = \mathbf{F}_{i,j}(\mathbf{S}(i,j,t), \theta) + [\mathcal{F}(i,j,t), 0]$$

I understand that the state $\mathbf{S}(i,j,t)$ at the grid point $i,j$ depends only on $\mathbf{S}(i,j,t)$ and the forcing. In order to avoid confusion, make the above equation depend on a defined global state, for example $\mathbf{S}(t) = [\mathbf{S}(0,0,t), \dots, \mathbf{S}(n_x, n_y, t)]^T \in \mathbb{R}^{3(n_x \times n_y)}$. Also, make sure the forcing notation is consistent with the definition of the global state vector $\mathbf{S}(t)$.

3. Regarding the optimization of the embedding parameters: after reading lines 347 to 352, I don't really understand how you parameterized your embedding parameters. In the appendix, it says that you use FNN to get the dimension, but it's not clear how you calibrate the delay. Please further explain your methodology here.

4. On the delay embedding parameterization: You state in lines 379 to 382 that in the Takens delay embedding theorem, there is no underlying assumption on the time delay. I have an example that contradicts this. Imagine that your limit-set is periodic with period T and that you use a time delay of T. In this situation, your reconstruction will not be diffeomorphic to the underlying limit set. Actually, this is written the other way around in the Fractal Delay Embedding Prevalence Theorem (Theorem 2.5 in [11]).

5. Benchmark: This is extremely important. You should provide a benchmark of your results to help readers better understand the performance of your work. At least compare to persistence, but I would recommend testing against other state-of-the-art machine learning models.

6. Presentation of the experiments: For the experimental section, I recommend including only the sensors configuration figure and a single forecasting figure per experiment. The remaining figures should be placed in the appendix.

7. On the dimension of the embedding: You state in lines 384 to 388 that the result of Sauer et al. indicates that if $D_E > 2D_A + 1$ the unprojection would work. I think that this result is from Whitney Embedding Prevalence Theorem (theorem 2.2 in [11]). Also, when you assume that $D_A$ is near 300 (lets say $D_A = 300$) then $D_E \approx 10, 20$ is very far from satisfying the condition $D_E > 2D_A + 1$. Please rectify this sentence.

8. On the scalability of the model: In your experiments, your initial state is of dimension 300. However, the embedding you used is of dimension $27 * 10$ to $27 * 20$. On higher dimensional systems as you mensioned, the dimension is much larger that 300 ($\mathbf{O}(10^6)$ for example). Do you expect your model to be in even larger dimensions than $\mathbf{O}(10^6)$. If it is the case, what is the cost of the model ? If not, why would you expect to get a smaller dimension.

Minor comments,

1. Do you mean embedding when you write unprojection ? If so, use embedding.

2. in lines 100 to 101, It's good to explain what one loses when bypassing equations.

3. in lines 96 to 97, I don't understand why you avoid uncertainties in the data. If your model is optimized from data and there is some uncertainty in the data, you would have uncertainty in the models.

4. I found the sentence from line 253 to 257 difficult to read.

5. Equation 6 has two commas.

I hope that the authors will understand my comments in a constructive way, and that I value their work and the time they invested in the preparation of the manuscript. It might be that I have misunderstood something, in this case, if something wasn't clear for me as a reviewer, it is possible that it wouldn't be clear also for the readers.

Response to Referee 2:
1) **The References [1-4] are excellent choices to add to the argument we make on the importance of regional weather models. References [5-10] provide a nice background as to what other methods exist out there and what they were used for. We have added them to our bibliography, thank you.**

2) **We've changed the notation in a way we hope pleases the referee and clarifies to future readers. The notation for r_1 and r_2 has been dropped in place of x and y reducing the total amount of notation in the paper. An additional line is included also to explain that the S(t) without any indication of location (i.e. r or (i,j)) is the full global set of state variables as suggested by the referee.**

3) **In our work using time delay embedding the method of FNN is used to estimate the dimension and the method of average mutual information is used to estimate the time delay. Using these estimates we then use grid search method, based on the quality of prediction. Grid Searching is used to fine tune the suggested values from these methods. We have added additional instruction on the use of the Average Mutual Information to the appendix.**

4) **This is a good mathematical point, and we thank the referee for raising it. If the attractor is a limit cycle with period T, then the Fourier spectrum will have a totally dominant frequency of 1/T, and that will be clear to the careful user, and the difficulty can be avoided. In the Taken Embedding Theorem, the theory itself makes no reference to the specifics of the necessary transformation for the state space to be properly reconstructed only that with infinitely accurate data the choice of time delay does not matter. However, since we never work with such data, It is as the referee has pointed out though that specific choices of transformations are import to the quality of the reconstruction for DDF's needs.**

5) **To bench mark our efforts we (in addition to cleaning up the graphs to make them more synergized and compact) include an error graph that accounts for an average percent error across all observed dimensions. This error graph as well as the results already reported should suffice to compare against similar papers on the study of reduced dimensional SWE in the use of data driven machine learning tasks [1-2]. Additionally, mentions of these papers are added to section 1.1 to reference them as comparable benchmark studies. Ultimately, this paper is focused on the study and application of DDF to an externally forced dynamical system; while the bench marking to Lorenz systems or other autonomous systems is interesting and useful for bench**

marking (something the authors have extensively and successfully experimented on), it goes beyond the scope of this paper as this would require a restructuring of the form of the DDF dynamics in a way that would be unfamiliar to the SWE DDF form and overly lengthen an already long paper. For these reasons we think a bench marking comparison with references [1-2] as listed serves this purpose best. We hope this answer is satisfactory to the referee.

6) **Thank you for this suggestion. We have combined the graphs relevant to each regional configuration with the graphic showing that configuration.**

7) **The criterion on $D_E$ relating it to the fractal dimension $D_A$ of the attractor tells us what the maximum $D_E$ might be. In practice, in many examples, $D_E$ is less than $2D_A + 1$. Take Lorenz63: $D_A \sim 2.06$, but $D_E$ is $3 < 5.12$. There may also be some confusion when we say $D_E = 10\text{-}20$ would work because we are saying that a total dimension of $27*10$ to $27*20$ would work as $D_E$ is the number of duplicates of the original observed data set.**

8) **The Whitney criterion is used in implementing the Takens embedding idea. $2D_A + 1$ is sufficient to capture motion on the attractor. If one uses a $D_E$ smaller than this, the observations are not yet unprojected. If one uses a dimension larger than this no harm is done.**
**We used the criterion of the quality of the forecast to choose both the time delay and $D_E$. This a different criterion than discussed by either Whitney or Takens. As long as $D_E > 2D_A + 1$, it is acceptable.**
**Heuristically, we've found that for good predictions fewer dimensions are needed than exist in the original data set, marginal improvement is made with the inclusion of extra and since we were working with a smaller model, we chose to spend the computational power to include more dimensions for less benefit (something that would most likely not be done with such large-scale models).**

Minor Comments
1) **We use unprojection in the context of false nearest neighbors when one has observed a subset of the complete set of state variables of the (unknown) source state space. Since the FNN method unprojects the observed data using the Physics information in the time delays, we argue that using "unprojection" conveys to the reader the action reason for the time delay construction.**
2) **We lose the information captured in the original construction of the model equations, specifically important things like the dynamical flow, the conservation laws, and other physics principles relevant to the data under study. We benefit though from DDF learning the natural rules itself from data on that exact physics though.**
3) **Working with data will always be noisy and have uncertainties in it, what we are saying here is that by using a data driven forecasting model, we are avoiding the extra uncertainties that may come along with using the prescribed models.**

4) **Thank you for suggesting we reword this. We have done so and attempted to be more clear than before.**

5) **Thank you for catching this. We have corrected it.**

References:

[1] Yıldız, Süleyman, et al. "Learning reduced-order dynamics for parametrized shallow water equations from data." *International Journal for Numerical Methods in Fluids* 93.8 (2021): 2803-2821.

[2] Chen, Xiaoqian, Balasubramanya T. Nadiga, and Ilya Timofeyev. "Predicting shallow water dynamics using echo-state networks with transfer learning." *GEM-International Journal on Geomathematics* 13.1 (2022): 20.

From The Editor:

Dear authors,

In addition to the two comments from the reviewers, I would like to highlight several points that will need to be considered for the next round:

1/ the title is a bit oversold (because you are applying the methodology to toy data, not real weather data), please rephrase it;

2/ the notations are sometimes too technical which makes reading difficult, please simplify them;

3/ the graphics are of poor quality and sometimes not very informative, please reduce their number and pay more attention to them;

4/ overall, there is a lack of synthesis and general clarity, please pay attention to this.
Sincerely,

Pierre Tandeo

Response to the Editor:
1) **We tweaked our title to more accurately reflect our work as requested by the editor:**
   **"Data Driven Regional Shallow Water Equation Forecasting"**
2) **We have made changes to the notation taking Referee 2's suggestions in hopes of making the paper more clear.**
3) **4) We rearranged our figures in a way that will hopefully clean up their position (we put them into a 2 by 2 grid with a u, v, and z graph with the fourth graph being an error estimate). We think these changes improve the overall clarity and give the readers more information on how DDF performs with the Regional SWE model.**

---

## Author Comment (AC2)

**UNIVERSITY OF CALIFORNIA, SAN DIEGO**
[Figure]

BERKELEY • DAVIS • IRVINE • LOS ANGELES • MERCED • RIVERSIDE • SAN DIEGO • SAN FRANCISCO      SANTA BARBARA • SANTA CRUZ

DEPARTMENT OF PHYSICS
MARINE PHYSICAL LABORATORY (MPL), Scripps Institution of Oceanography

9500 GILMAN DRIVE
LA JOLLA, CALIFORNIA 92093-0357

07 February 2023

To: Dr. Pierre Tando, Editor ***Nonlinear Processes in Geophysics***

Re; Manuscript: *"Data Driven Regional Weather Forecasting : Example using the Shallow Water Equations."*

Dear Dr. Tando,

We submit an extensively revised manuscript for this paper along with detailed responses to the two referees.

The issues presented by the two reviewers have been addressed in detail. We have added a new section on a benchmark evaluation of the error in the calculations discussed in the paper. It is Section 6 and the regional error metric is Equation (19) in that section. Section 6 also adds Figure 10 where the regional error as a function of time in the forecasting window is presented for each of the scenarios we discuss.

We also combined our results in each of the scenarios resulting in a reduction of the number of figures from 22 to 9 in the existing sections of the paper.

We express our appreciation to the two reviewers and to you for assisting us in improving the manuscript.

Yours sincerely,

Luke Fairbanks on behalf of Randall Clark, Ramon Sanchez, Pacharadech Wacharanan, and Henry Abarbanel